# Exploring association between place of delivery and newborn care with early-neonatal mortality in Bangladesh

**Rashida-E Ijdi**[1,2]*, **Katherine Tumlinson**[1,2], **Siân L. Curtis**[1,2]

**1** Department of Maternal and Child Health, Gillings School of Global Public Health, The University of North Carolina at Chapel Hill, Chapel Hill, North Carolina, United States of America, **2** Carolina Population Center, The University of North Carolina at Chapel Hill, Chapel Hill, North Carolina, United States of America

☯ These authors contributed equally to this work.
* rashida.ijdi@gmail.com

**Data Availability Statement:** The data sources used in the analysis can be accessed at the web link upon request https://www.dhsprogram.com/data/available-datasets.cfm.

## Abstract

### Objective

Bangladesh achieved the fourth Millennium Development Goal well ahead of schedule, with a significant reduction in under-5 mortality between 1990 and 2015. However, the reduction in neonatal mortality has been stagnant in recent years. The purpose of this study is to explore the association between place of delivery and newborn care with early neonatal mortality (ENNM), which represents more than 80% of total neonatal mortality in Bangladesh.

### Methods

In this study, 2014 Bangladesh Demographic and Health Survey data were used to assess early neonatal survival in children born in the three years preceding the survey. The roles of place of the delivery and newborn care in ENNM were examined using multivariable logistic regression models adjusted for clustering and relevant socio-economic, pregnancy, and newborn characteristics.

### Results

Between 2012 and 2014, there were 4,624 deliveries in 17,863 sampled households, 39% of which were delivered at health facilities. The estimated early neonatal mortality rate during this period was 15 deaths per 1,000 live births. We found that newborns who had received at least 3 components of essential newborn care (ENC) were 56% less likely to die during the first seven days of their lives compared to their counterparts who received 0–2 components of ENC (aOR: 0.44; 95% CI: 0.24–0.81). In addition, newborns who had received any postnatal care (PNC) were 68% less likely to die in the early neonatal period than those who had not received any PNC (aOR: 0.32; 95% CI: 0.16–0.64). Facility delivery was not significantly associated with the risk of early newborn death in any of the models.

**Funding:** The author(s) received no specific funding for this work.

**Competing interests:** The authors have declared that no competing interests exist.

**Abbreviations:** ANC, antenatal care; aOR, adjusted odds ratio; BDHS, Bangladesh demographic and health survey; BEmONC, basic emergency obstetric and neonatal care; BHFS, Bangladesh health facility survey; CEmONC, comprehensive emergency obstetric and neonatal care; CI, confidence interval; ENNM, early neonatal mortality; ENC, essential newborn care; PNC, postnatal care; SDG, sustainable development goals; SOP, standard operating procedure; VIF, variance inflation factors; WHO, World Health Organization.

## Conclusion

Our study findings highlight the importance of newborn and postnatal care in preventing early neonatal deaths. Further, findings suggest that increasing the proportion of women who give birth in a healthcare facility is not sufficient to reduce ENNM by itself; to realize the theoretical potential of facility delivery to avert neonatal deaths, we must also ensure quality of care during delivery, guarantee all components of ENC, and provide high-quality early PNC. Therefore, sustained efforts to expand access to high-quality ENC and PNC are needed in health facilities, particularly in facilities serving low-income populations.

## Introduction

Early neonatal mortality (ENNM), defined as the death of a newborn between zero and six days after birth, represents 73% of all neonatal deaths (i.e., deaths occurring during 0–27 days of life) worldwide [1]. As such, the early neonatal period is the most vulnerable time for a child's survival. ENNM reflects the quality of care received by the mother during the antenatal period and at childbirth [2, 3]. The availability, accessibility and quality of skilled birth attendants and emergency obstetric care are considered important for reducing the burden of early-neonatal deaths [3, 4]. Globally, the neonatal mortality rate fell by 49%, from 37 deaths in 1990 to 18 deaths per 1,000 live births in 2017, but that decline was slower than the decline in mortality among children aged 1–59 months [5]. Almost 99% of neonatal deaths occur in low and middle-income countries [6].

There has been a steady downward trend in childhood mortality in Bangladesh, with a 46% decline in neonatal mortality and a 65% decline in under-five mortality [7] over the last two decades. As a result, Bangladesh achieved the fourth Millennium Development Goal target ahead of schedule [8]. As the overall rates of child mortality have decreased, deaths have become increasingly concentrated in the earliest month of life, especially in the early-neonatal period; ENNM as a percent of all infant mortality increased from 30% of all infant deaths in 1993–1994 to 58% in 2010–2014 [7]. To achieve the United Nations' Sustainable Development Goal (SDG) target 3.2 (i.e., reduction of neonatal deaths to at least as low as 12 per 1,000 live births), Bangladesh needs to reduce neonatal deaths by 57% [9].

An important step for low- and middle-income countries in meeting the SDG goal to reduce neonatal mortality is to provide universal access to skilled care at birth [10, 11]. To increase access to skilled birth attendants, countries with a high proportion of home deliveries (including Bangladesh) are attempting to move all deliveries out of the home to health institutions (hereafter referred to as health facilities) [9]. More than 40% of maternal deaths, stillbirths and early-neonatal deaths occur each year during delivery [11]. Thus, delivery in a health facility with a skilled provider has been shown to reduce ENNM in some contexts [12–14]. As use of facility delivery increases in Bangladesh, ensuring a high level of essential newborn care (ENC) in health facilities will also be vital in maintaining the momentum to decrease neonatal mortality [15, 16].

However, the relationship between delivery care and maternal and early neonatal mortality can be complex in practice. A study in Bangladesh showed that maternal and ENNM rates were much higher among women delivering in a health facility, especially in a higher-level facility, than among those delivering at home [17]. When uptake of skilled birth attendance or comprehensive obstetric care is low, women will often only seek skilled care when their labor

becomes complicated, and they may do so too late for a midwife or doctor to be able to save the lives of the mothers or neonates [17]. Notably, this difference in mortality between home and facility birth decreased as the percentage seeking skilled delivery care increased over time [18]. It is assumed that facility delivery confirms the ENC components and ensures postnatal check-up within 48 hours of delivery, which positively impacts delivery outcomes and thereby reduces the risks of ENNM. Unfortunately, it is difficult to obtain valid and comparable data on complications for home and facility deliveries in order to ascertain the true causal relationship between place of delivery and ENNM in Bangladesh. Even in high-income settings, it is impossible to capture all risk factors for complications that direct women towards choosing a facility delivery [18].

It is important to explore the observed association between place of delivery and ENNM in Bangladesh to gain insights into how changes in facility deliveries might or might not be associated with changes in ENNM. Studies suggest that better monitoring and management of labor, delivery, and postpartum are critical for reducing maternal and perinatal mortality (i.e., a stillbirth or early neonatal death) [19–21]. This study aims to explore the association of the place of delivery and newborn care with ENNM in Bangladesh. Despite high ENNM in Bangladesh, few studies have been carried out to date focusing on these issues, and most of them were small-scale and clinically oriented [15, 17, 22–25]. To our knowledge, this is the largest cross-sectional study to estimate the association between place of delivery or newborn care and ENNM in Bangladesh. This information will assist policymakers and program managers in the health sector in formulating appropriate strategies and interventions to provide high-quality health services and interventions to further improve newborn and maternal health.

## Methodology

### Conceptual framework

In this paper, the conceptual framework for ENNM is guided by Mosley and Chen's framework for studying child survival in low- and middle-income countries [26]. This framework seeks to identify factors that directly and indirectly impact the outcome of interest. Our primary outcome is ENNM, which comprises deaths within the first seven days (0–6 days) of life. Potential risk factors for ENNM are depicted in the conceptual framework shown in Fig 1.

The modified conceptual framework groups the proximate determinants into three categories: maternal, pregnancy, and early-neonatal factors. While several distal determinants could be considered for our analysis, for the purpose of this study we chose to focus on two socio-demographic factors (viz., household wealth quintile and place of residence -urban or rural) and one community factor (viz., time to reach to the nearest health facility, which is a proxy for distance to the nearest health facility) for this study.

In specifying our model, we have also included a directed acyclic graph (DAG) [27, 28] as an annex, to identify potential confounders and select a minimally sufficient adjustment set [29, 30].

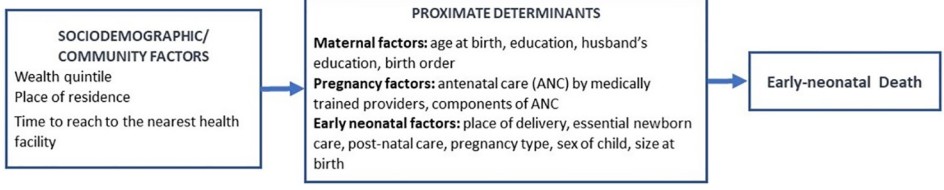

**Fig 1. Conceptual framework of early neonatal mortality.**

## Data source

The study uses data from the 2014 Bangladesh Demographic and Health Survey (BDHS)- a nationally representative cross-sectional survey using a two-stage stratified random sampling of households, which covered all districts and administrative divisions of Bangladesh. The detailed methodology can be found in the 2014 BDHS report [7]. The survey collected information from 17,863 ever-married women ages 15–49 years, who were asked to provide a complete history of their live births. The BDHS collected information on pregnancy and delivery care for the most recent live births in the three years preceding the survey. Therefore, we carried out our final analysis using the 4,624 live births, who were the last birth born in the three years preceding the survey. If the last recent birth was a multiple birth only the youngest of the multiple births was included in the sample because data on ENC and postnatal care was only collected for that child. In this final sample we observed 69 deaths during the early neonatal period.

## Data analysis

**Outcome variable.**   Our outcome variable was ENNM, defined as the death of a live-born child within the first seven days of life (0–6 days).

**Exposure variables.**   Our primary exposure variable was the place of delivery, categorized as home or facility delivery. Facility delivery was defined as births occurring in any health facility including hospitals, health centers, or clinics. Home delivery was defined as giving birth to a child in a residence. We hypothesize that mothers who delivered their children in a facility are less likely to lose their newborns in the first week of life.

ENC and any infant postnatal care (PNC) were secondary exposure variables in this study. DHS defines an infant as having received any PNC if they receive PNC within 41 days after delivery–included as child PNC. The ENC variable is a composite indicator derived from the components of ENC measured by the BDHS: use of sterilized instruments to cut the umbilical cord, applying nothing to the cord, immediate drying (within 5 minutes) by keeping the baby warm, delaying bathing to 72 hours after birth, and initiating breastfeeding within 1 hour of delivery. Each component was weighted equally, and a score was developed for the number of ENC components received by the infant (0–5). The score was then split at the middle of the categories and recoded into a binary ENC variable for low (0–2 components) or high (3–5 components) of ENC.

**Covariates.**   Based on our review of the literature, we recognize that the relationship between place of delivery, as the primary exposure, and ENNM, as the outcome, may be confounded by several factors, as depicted in the DAG in the annex, and therefore include these covariates in our model. Covariates included for socio-demographic/community factors (residence, wealth, time to reach the nearest health facility) maternal factors (age, birth order, maternal education, paternal education), pregnancy factors (antenatal care (ANC) attended by a medically trained provider, and receipt of ANC components), and neonatal factors (pregnancy type, sex and size of the child at birth). All the covariates were defined as categorical variables for the analysis. ANC components included: blood-pressure taken, urine tested, blood sample taken, weight taken, ultrasonogram conducted, and discussion of danger signs at least once during any of the ANC visits. Each component was weighted equally, and a score was constructed for the number of ANC components received during the pregnancy (0–6). The score was recoded into a composite categorical ANC variable with categories none, 1–3, 4–5 or all 6 ANC components so that each of the categories has a similar number of observations. Further information on the study variables can be found in the 2014 BDHS report [7].

## Statistical analysis

Descriptive analysis was performed to examine the distribution of the variables. We conducted bivariate analysis to test the association between each of the independent variables in the conceptual framework with the primary exposure variable—place of delivery (home vs. facility) using the Rao-Scott test to account for the complex survey design.

Multivariable logistic regression models were used to calculate odds ratios (ORs) and 95% confidence intervals (CIs) to assess the odds of ENNM by place of delivery, ENC and PNC. Crude ORs were initially calculated to examine the bivariate relationship between the covariates and ENNM. Adjusted ORs (aOR) were then calculated in two final models. In the first of these models, socio-demographic/community, maternal, pregnancy, and neonatal factors were added to the model with delivery care to determine how the association between delivery care and ENNM is affected by controlling for these potential confounders. In the second model, the two secondary exposure variables of interest, ENC and PNC, are added to the model. Given the large number of covariates, multicollinearity was assessed using variance inflation factors (VIFs). All covariates were acceptable with VIFs less than 5 (our calculated VIF was 1.77). Since correlations between variables were not a concern, all covariates were retained in the final model. All analysis was performed using statistical software package STATA version 16.0 [31], and appropriate sampling weights for BDHS 2014 were applied using Stata's survey estimation procedures ("svy" command) to achieve nationally representative estimates of the population of Bangladesh after adjusting for sample strata and clusters. Standard significance levels are reported (i.e., no correction for multiple comparisons). All the analyses were conducted using survey weights and accounted for clustering.

**Ethics approval and consent to participate.** The Bangladesh Demographic and Health Survey (BDHS) 2014 data are public access data and were made available to us by MEASURE DHS (https://dhsprogram.com/) upon request. BDHS data collection procedures were approved by the Institutional Review Board (IRB) of the ICF International, Rockville, MD, USA, and the Ethical Review Committee of Bangladesh Medical Research Council (BMRC), Dhaka, Bangladesh. ICF IRB ensures that the survey complies with the U.S. Department of Health and Human Services regulations for the protection of human subjects (45 CFR 46), while the no-objection from BMRC ensures that the survey complies with laws and norms of the country. Before each interview, an informed consent statement was read to the respondent, who might accept or decline to participate. The BDHS informed consent statements provided details on the aim and procedure of the survey, potential risks and benefits to the respondent, and contact information for a person who can provide the respondent with more information about the interview. Most importantly, the informed consent statement emphasized that participation was voluntary; that the respondent might refuse to answer any question, or terminate participation at any time, and that the respondent's identity and information would be kept strictly confidential. Details of the ethical approval can be found elsewhere [7]. Ethical approval for this research was waived by the authors' institute, i.e., UNC Chapel Hill Office of Human Research Ethics/IRB, because this study was carried out using publicly available data that are anonymized and free of personally identifiable information.

## Results

Table 1 describes the background characteristics of live births in the three years preceding the survey by ENNM status (i.e., death or survival). Out of all births in our sample, 39% were delivered at health facilities, and 40% ENNM occurred at health facilities. Low ENC (0–2 components) comprise 30% of all births and 50% for ENNM. And, 37% neonates received no PNC and 59% of ENNM. Male births comprise 53% of all births and 54% of ENNM. The majority

**Table 1. Distribution of most recent live births in the three years before the survey by background characteristics and neonatal survival status, Bangladesh DHS 2014.**

| Background characteristics | Number of early neonatal deaths[1] (n = 69) | Number of children survived from early neonatal period (n = 4,555) | Number of births (n = 4,624) |
|---|---|---|---|
| | n (%) | n (%) | n (%) |
| **Primary variables of interest** | | | |
| **Place of delivery** | | | |
| Home | 42 (60.1) | 2,798 (61.4) | 2,840 (61.4) |
| Health Facility | 27 (39.9) | 1,757 (38.6) | 1,784 (38.6) |
| **Components of Essential Newborn Care (ENC)** | | | |
| Low ENC (0–2 components) | 35 (50.1) | 1,370 (30.1) | 1,405 (30.4) |
| High ENC (3 or more components) | 34 (49.9) | 3,185 (69.9) | 3,219 (69.6) |
| **Child Postnatal Care (PNC)** | | | |
| Received no PNC | 41 (59.1) | 1,652 (36.3) | 1,693 (36.6) |
| Received PNC | 28 (40.9) | 2,903 (63.7) | 2,931 (63.4) |
| **Neonatal characteristics** | | | |
| **Pregnancy type** | | | |
| Twin | 5 (7.6) | 24 (0.5) | 29 (0.6) |
| Singleton | 64 (92.4) | 4,531 (99.5) | 4,595 (99.4) |
| **Sex of the child** | | | |
| Male | 37 (54.2) | 2,400 (52.7) | 2,437 (52.7) |
| Female | 32 (45.8) | 2,155 (47.3) | 2,187 (47.3) |
| **Size of child at birth** | | | |
| Very small or small | 17 (24.7) | 905 (19.9) | 922 (20.0) |
| Average or larger | 52 (75.3) | 3,650 (80.1) | 3,702 (80.0) |
| **Pregnancy care characteristics** | | | |
| **Antenatal Care (ANC) check up** | | | |
| ANC by non-MTP[2]/no ANC | 25 (35.8) | 1,586 (34.8) | 1,611 (34.8) |
| ANC by MTP[2] | 44 (64.2) | 2,969 (65.2) | 3,013 (65.2) |
| **ANC components** | | | |
| No components | 18 (26.4) | 995 (21.8) | 1,013 (21.9) |
| 1–3 components | 17 (24.1) | 1,191 (26.1) | 1,208 (26.1) |
| 4–5 components | 15 (22.4) | 1,356 (29.8) | 1,371 (29.7) |
| All 6 components | 19 (27.1) | 1,013 (22.2) | 1,032 (22.3) |
| **Sociodemographic characteristics** | | | |
| **Mother's age at birth** | | | |
| <20 | 25 (35.9) | 1,447 (31.8) | 1,472 (31.8) |
| 20–34 | 41 (59.4) | 2,927 (64.3) | 2,968 (64.2) |
| 35+ | 3 (4.7) | 181 (3.9) | 184 (4.0) |
| **Birth order** | | | |
| 1 | 32 (46.9) | 1,812 (39.8) | 1,844 (39.9) |
| 2–4 | 34 (48.4) | 2,443 (53.6) | 2,477 (53.5) |
| 5+ | 3 (4.7) | 300 (6.6) | 303 (6.6) |
| **Maternal educational attainment** | | | |
| No formal education | 6 (9.3) | 647 (14.2) | 653 (14.1) |
| Primary incomplete | 13 (19.1) | 736 (16.1) | 749 (16.2) |
| Primary completed | 10 (14.7) | 534 (11.7) | 544 (11.8) |
| Secondary incomplete | 34 (49.0) | 1,857 (40.8) | 1,891 (40.9) |
| Secondary completed or higher | 6 (7.9) | 781 (17.2) | 787 (17.0) |

*(Continued)*

**Table 1.** (Continued)

| Background characteristics | Number of early neonatal deaths[1] (n = 69) | Number of children survived from early neonatal period (n = 4,555) | Number of births (n = 4,624) |
|---|---|---|---|
| | n (%) | n (%) | n (%) |
| **Paternal educational attainment** | | | |
| No formal education | 19 (28.0) | 1,084 (23.8) | 1,103 (23.8) |
| Primary incomplete | 16 (22.7) | 737 (16.2) | 753 (16.3) |
| Primary completed | 9 (13.1) | 624 (13.7) | 633 (13.7) |
| Secondary incomplete | 16 (23.7) | 1,138 (25.0) | 1,155 (25.0) |
| Secondary completed or higher | 9 (12.5) | 972 (21.3) | 980 (21.2) |
| **Wealth quintile** | | | |
| Poorest | 22 (32.2) | 979 (21.5) | 1,001 (21.7) |
| Poorer | 14 (19.8) | 861 (18.9) | 875 (18.9) |
| Middle | 10 (14.7) | 872 (19.1) | 882 (19.1) |
| Richer | 13 (19.1) | 941 (20.7) | 954 (20.6) |
| Richest | 10 (14.2) | 902 (19.8) | 912 (19.7) |
| **Place of residence** | | | |
| Rural | 53 (77.3) | 3,362 (73.8) | 3,415 (73.9) |
| Urban | 16 (22.7) | 1,193 (26.2) | 1,209 (26.1) |
| **Community level characteristics** | | | |
| **Time to reach to the nearest health facility** | | | |
| 0–15 minutes | 38 (54.7) | 2,383 (52.3) | 2,421 (52.4) |
| 16–30 minutes | 27 (39.0) | 1,727 (37.9) | 1,754 (37.9) |
| More than 30 minutes | 4 (6.3) | 445 (9.8) | 449 (9.7) |

[1] Early neonatal deaths were deaths at age 0–6 days among live-born children.

[2] Medically trained provider (MTP) includes: qualified doctor, nurse/midwife/paramedic, family welfare visitor (FWV), community skilled birth attendant (CSBA), and sub-assistant community medical officer (SACMO).

of births (64%) and deaths (59%) are to women aged 20–34. Births to women in the poorest wealth quintile comprise 22% of births but 32% of deaths, while 20% of births and 14% of deaths are among women in the richest wealth quintile. Around three quarters of births and deaths occur in rural areas.

Table 2 presents early neonatal outcomes by place of delivery. Overall, 1,784 live births (39%) occurred in a health facility, and 2,840 live births (61%) occurred in a home setting. A total of 69 early neonatal deaths occurred during the study timeframe, for an overall ENNM rate of 14.9 deaths per 1,000 live births. Among all ENN deaths, almost half (46%) of the early neonatal deaths occurred within 24 hours after delivery. The ENNM rate was slightly higher at health facilities (15.7) than for home deliveries (14.4).

**Table 2. Early neonatal outcomes (in the three years preceding the survey) by place of delivery, Bangladesh DHS 2014.**

| | Home | Health Facility | Number of births |
|---|---|---|---|
| | n (%) | n (%) | n |
| Total number of deliveries | 2,840 (61.4) | 1,784 (38.6) | 4,624 |
| **Early neonatal deaths** | **41 (60.1)** | **28 (39.9)** | **69** |
| Early neonatal deaths in 24 hours after delivery | 23 (73.0) | 9 (27.0) | 32 |
| Early neonatal deaths on days '1–6' after birth | 18 (49.3) | 19 (50.7) | 37 |
| **Early Neonatal Mortality rate (per 1,000 live births)** | **14.4** | **15.7** | **14.9** |

The proportion of deliveries occurring in health facilities varies significantly according to several variables included in our analysis, as presented in Table 3. Women who had not received antenatal care from medically trained providers mostly delivered at home (83%), and the percentage of women who delivered in a health facility increases as the number of ANC

**Table 3. Distribution of place of delivery by neonatal, pregnancy care, maternal, and socio-demographic/community characteristics among most recent live births in the three years preceding the survey, Bangladesh DHS 2014.**

| | Home (n = 2,840) | Health Facility (n = 1,784) | Total (n = 4,624) | p-value[1] |
|---|---|---|---|---|
| | n (%) | n (%) | n | |
| **Neonatal characteristics** | | | | |
| **Pregnancy type** | | | | |
| Twin | 13 (46.3) | 16 (53.7) | 29 | 0.139 |
| Singleton | 2,827 (61.5) | 1,768 (38.5) | 4,595 | |
| **Sex of child** | | | | |
| Male | 1,468 (60.2) | 969 (39.8) | 2,438 | 0.123 |
| Female | 1,372 (62.7) | 815 (37.3) | 2,186 | |
| **Size of child at birth** | | | | |
| Small/very small | 600 (65.1) | 322 (34.9) | 922 | 0.076 |
| Average or larger | 2,240 (60.5) | 1,462 (39.5) | 3,702 | |
| **Pregnancy care characteristics** | | | | |
| **Antenatal Care (ANC) check up** | | | | |
| ANC by non-MTP[2]/no ANC | 1,330 (82.6) | 281 (17.4) | 1,611 | <0.001 |
| ANC by MTP[2] | 1,510 (50.1) | 1,503 (49.9) | 3,012 | |
| **ANC components** | | | | |
| No components | 893 (88.1) | 121 (11.9) | 1,013 | <0.001 |
| 1–3 components | 887 (73.5) | 320 (26.5) | 1,207 | |
| 4–5 components | 758 (55.3) | 613 (44.7) | 1,372 | |
| All 6 components | 302 (29.3) | 730 (70.7) | 1,032 | |
| **Sociodemographic characteristics** | | | | |
| **Maternal age at birth** | | | | |
| <20 | 921 (62.6) | 551 (37.4) | 1,472 | 0.371 |
| 20–34 | 1,796 (60.5) | 1,172 (39.5) | 2,968 | |
| 35+ | 123 (66.9) | 61 (33.1) | 184 | |
| **Birth order** | | | | |
| 1 | 957 (51.9) | 888 (48.1) | 1,845 | <0.001 |
| 2–4 | 1,625 (65.6) | 851 (34.4) | 2,476 | |
| 5+ | 258 (85.3) | 45 (14.7) | 303 | |
| **Maternal educational attainment** | | | | |
| No formal education | 548 (84.0) | 105 (16.0) | 653 | <0.001 |
| Primary incomplete | 571 (76.2) | 178 (23.8) | 749 | |
| Primary completed | 378 (69.5) | 166 (30.5) | 544 | |
| Secondary incomplete | 1,101 (58.2) | 790 (41.8) | 1,891 | |
| Secondary completed or higher | 242 (30.7) | 545 (69.3) | 787 | |
| **Paternal educational attainment** | | | | |
| No formal education | 885 (80.2) | 219 (19.8) | 1,103 | <0.001 |
| Primary incomplete | 549 (72.9) | 204 (27.1) | 753 | |
| Primary completed | 425 (67.2) | 208 (32.8) | 633 | |
| Secondary incomplete | 652 (56.4) | 503 (43.6) | 1,155 | |
| Secondary completed or higher | 330 (33.7) | 650 (66.3) | 980 | |

*(Continued)*

**Table 3.** (Continued)

| | Home (n = 2,840) | Health Facility (n = 1,784) | Total (n = 4,624) | p-value[1] |
|---|---|---|---|---|
| | n (%) | n (%) | n | |
| **Wealth Quintile** | | | | |
| Poorest | 848 (84.7) | 153 (15.3) | 1,001 | <0.001 |
| Poorer | 660 (75.5) | 215 (24.5) | 875 | |
| Middle | 575 (65.2) | 307 (34.8) | 882 | |
| Richer | 499 (52.4) | 455 (47.6) | 954 | |
| Richest | 258 (28.3) | 654 (71.7) | 912 | |
| **Place of residence** | | | | |
| Rural | 2,336 (68.4) | 1,079 (31.6) | 3,415 | <0.001 |
| Urban | 504 (41.7) | 705 (58.3) | 1,209 | |
| **Community level characteristics** | | | | |
| **Time to reach to the nearest health facility** | | | | |
| 0–15 minutes | 1,297 (53.6) | 1,124 (46.4) | 2,421 | <0.001 |
| 15–30 minutes | 1,183 (67.5) | 571 (32.5) | 1,754 | |
| More than 30 minutes | 360 (80.2) | 89 (19.8) | 449 | |

[1] Rao-Scott test.

[2] Medically trained provider (MTP) includes: qualified doctor, nurse/midwife/paramedic, family welfare visitor (FWV), community skilled birth attendant (CSBA), and sub-assistant community medical officer (SACMO).

components received increases. Facility delivery was lower with higher birth order and higher as both maternal and paternal level of educational attainment increased. Women who lived in rural areas were more likely to deliver at home (68%) than women in urban areas (42%). Facility delivery also increases with increasing wealth. Almost 46% of women delivered at a health facility when the time to reach the nearest facility was 0–15 minutes compared to 20% of women who lived more than 30 minutes from a health facility.

Table 4 shows the percentage of births that received ENC and PNC by place of delivery for the study sample. Around three quarters of neonates who were delivered at health facilities received high ENC (3 or more components) compared with two-third (66%) among the neonates born at home. A little over half of neonates delivered at home received any PNC compared with 82% of neonates delivered at a health facility.

Results of the logistic regression models are presented in Table 5, where model 1 is the crude or unadjusted model, model 2 adjusts the effect of delivery care for the potential

**Table 4. Distribution of essential newborn care and postnatal care by place of delivery among most recent live births in the three years preceding the survey, Bangladesh DHS 2014.**

| | Home (n = 2,840) | Health Facility (n = 1,784) | Total (n = 4,624) | p-value[1] |
|---|---|---|---|---|
| | n (%) | n (%) | n (%) | |
| **Newborn care characteristics** | | | | |
| **Components of Essential Newborn Care (ENC)** | | | | |
| Low ENC (0–2 components) | 973 (34.3) | 432 (24.2) | 1,405 (30.4) | <0.001 |
| High ENC (3 or more components) | 1,867 (65.7) | 1,352 (75.8) | 3,219 (69.6) | |
| **Child Postnatal Care (PNC)** | | | | |
| Received no PNC | 1,374 (48.4) | 320 (17.9) | 1,693 (36.6) | <0.001 |
| Received any PNC | 1,466 (51.6) | 1,464 (82.1) | 2,931 (63.4) | |

[1] Rao-Scott test.

**Table 5. Logistic regression estimates of the association of early neonatal mortality with delivery care, essential newborn care, and child postnatal care controlling for neonatal, pregnancy care, sociodemographic and community characteristics, Bangladesh DHS 2014.**

| | Unadjusted Model 1 (n = 4,624) | P value | Adjusted Model 2 (n = 4,624) | P value | Adjusted Model 3 (n = 4,624) | P value |
|---|---|---|---|---|---|---|
| | OR (95% CI) | | OR (95% CI) | | OR (95% CI) | |
| **Primary variables of interest** | | | | | | |
| **Place of delivery** | | | | | | |
| Home | 1.00 | - | 1.00 | - | 1.00 | - |
| Health Facility | 1.05 (0.61–1.80) | 0.840 | 1.09 (0.58–2.04) | 0.768 | 1.49 (0.43–1.96) | 0.230 |
| **Components of Essential Newborn Care (ENC)** | | | | | | |
| Low ENC (0–2 components) | 1.00 | - | | | 1.00 | - |
| High ENC (3 or more components) | 0.42 (0.24–0.75) | 0.003 | | | 0.44 (0.24–0.81) | 0.009 |
| **Child Postnatal Care (PNC)** | | | | | | |
| Received no PNC | 1.00 | - | | | 1.00 | - |
| Received PNC | 0.39 (0.21–0.71) | 0.003 | | | 0.32 (0.16–0.64) | 0.001 |
| **Neonatal characteristics** | | | | | | |
| **Pregnancy Type** | | | | | | |
| Twin | 1.00 | - | 1.00 | - | 1.00 | - |
| Singleton | 0.06 (0.02–0.17) | <0.001 | 0.05 (0.01–0.17) | <0.001 | 0.05 (0.01–0.16) | <0.001 |
| **Sex of the child** | | | | | | |
| Male | 1.00 | - | 1.00 | - | 1.00 | - |
| Female | 0.94 (0.53–1.65) | 0.833 | 0.86 (0.48–1.53) | 0.628 | 0.85 (0.47–1.53) | 0.595 |
| **Size of child at birth[1]** | | | | | | |
| Very small or small | 1.00 | - | 1.00 | | 1.00 | - |
| Average or larger | 0.75 (0.37–1.51) | 0.431 | 0.88 (0.42–1.82) | 0.736 | 0.92 (0.43–1.96) | 0.849 |
| **Pregnancy care characteristics** | | | | | | |
| **ANC by Medically trained Provider (MTP)** | | | | | | |
| ANC by non-MTP/no ANC | 1.00 | - | 1.00 | - | 1.00 | - |
| ANC by MTP | 0.95 (0.53–1.71) | 0.881 | 1.35 (0.48–3.77) | 0.558 | 1.56 (0.55–4.43) | 0.402 |
| **ANC Components** | | | | | | |
| No components | 1.00 | - | 1.00 | - | 1.00 | - |
| 1–3 components | 0.76 (0.36–1.60) | 0.476 | 0.61 (0.22–1.63) | 0.325 | 0.66 (0.23–1.91) | 0.438 |
| 4–5 components | 0.62 (0.27–1.38) | 0.246 | 0.57 (0.14–2.30) | 0.432 | 0.65 (0.15–2.86) | 0.569 |
| All 6 components | 1.00 (0.45–2.21) | 0.986 | 0.94 (0.23–3.86) | 0.942 | 1.22 (0.28–5.17) | 0.788 |
| **Sociodemographic characteristics** | | | | | | |
| **Mother's age at birth** | | | | | | |
| <20 | 1.00 | - | 1.00 | - | 1.00 | - |
| 20–34 | 0.81 (0.44–1.51) | 0.522 | 1.38 (0.68–2.83) | 0.365 | 1.49 (0.71–3.12) | 0.288 |
| 35+ | 1.03 (0.29–3.70) | 0.953 | 2.71 (0.39–18.49) | 0.307 | 2.77 (0.42–18.00) | 0.283 |
| **Birth order** | | | | | | |
| 1 | 1.00 | - | 1.00 | - | 1.00 | - |
| 2–4 | 0.76 (0.42–1.36) | 0.367 | 0.50 (0.25–0.99) | 0.047 | 0.49 (0.24–1.00) | 0.051 |
| 5+ | 0.60 (0.16–2.21) | 0.450 | 0.23 (0.02–2.40) | 0.221 | 0.26 (0.02–2.39) | 0.236 |
| **Maternal educational attainment** | | | | | | |
| No formal education | 0.54 (0.22–1.34) | 0.186 | 0.40 (0.14–1.17) | 0.097 | 0.38 (0.12–1.17) | 0.094 |
| Primary incomplete | 0.98 (0.44–2.16) | 0.962 | 0.68 (0.27–1.70) | 0.415 | 0.68 (0.26–1.78) | 0.440 |
| Primary completed | 1.04 (0.41–2.58) | 0.930 | 0.80 (0.30–2.12) | 0.664 | 0.85 (0.33–2.18) | 0.736 |
| Secondary incomplete | 1.00 | - | 1.00 | - | 1.00 | - |
| Secondary completed or higher | 0.38 (0.13–1.08) | 0.071 | 0.42 (0.11–0.55) | 0.195 | 0.43 (0.11–1.58) | 0.205 |
| **Paternal educational attainment** | | | | | | |

*(Continued)*

**Table 5.** (Continued)

| | Unadjusted Model 1 (n = 4,624) | P value | Adjusted Model 2 (n = 4,624) | P value | Adjusted Model 3 (n = 4,624) | P value |
| --- | --- | --- | --- | --- | --- | --- |
| | OR (95% CI) | | OR (95% CI) | | OR (95% CI) | |
| No formal education | 1.23 (0.53–2.83) | 0.617 | 1.54 (0.65–3.67) | 0.319 | 1.53 (0.64–3.64) | 0.328 |
| Primary incomplete | 1.47 (0.62–3.49) | 0.377 | 1.62 (0.66–3.98) | 0.286 | 1.56 (0.62–3.93) | 0.335 |
| Primary completed | 1.01 (0.36–2.75) | 0.987 | 1.06 (0.41–2.76) | 0.889 | 1.10 (0.40–2.99) | 0.842 |
| Secondary incomplete | 1.00 | - | 1.00 | - | 1.00 | - |
| Secondary completed or higher | 0.61 (0.21–1.73) | 0.359 | 0.79 (0.25–2.48) | 0.690 | 0.79 (0.25–2.45) | 0.687 |
| **Place of residence** | | | | | | |
| Rural | 1.00 | - | 1.00 | - | 1.00 | - |
| Urban | 0.82 (0.44–1.53) | 0.550 | 0.89 (0.45–1.76) | 0.742 | 0.89 (0.44–1.79) | 0.763 |
| **Wealth quintile** | | | | | | |
| Poorest | 1.00 | - | 1.00 | - | 1.00 | - |
| Poorer | 0.69 (0.30–1.57) | 0.387 | 0.58 (0.25–1.37) | 0.219 | 0.56 (0.23–1.34) | 0.196 |
| Middle | 0.51 (0.21–1.21) | 0.129 | 0.45 (0.19–1.03) | 0.060 | 0.44 (0.18–0.03) | 0.060 |
| Richer | 0.61 (0.27–1.37) | 0.238 | 0.52 (0.21–1.30) | 0.165 | 0.56 (0.22–1.45) | 0.241 |
| Richest | 0.47 (0.20–1.13) | 0.096 | 0.40 (0.13–1.19) | 0.103 | 0.45 (0.15–1.35) | 0.158 |
| **Community level characteristics** | | | | | | |
| **Time to reach the nearest health facility** | | | | | | |
| 0–15 minutes | 1.00 | - | 1.00 | - | 1.00 | - |
| 16–30 minutes | 0.98 (0.54–1.76) | 0.951 | 0.73 (0.40–1.33) | 0.313 | 0.74 (0.40–1.38) | 0.354 |
| More than 30 minutes | 0.62 (0.19–1.98) | 0.421 | 0.50 (0.15–1.61) | 0.248 | 0.44 (0.13–1.51) | 0.194 |

newborn, pregnancy care, maternal, and socio-demographic/community confounders, and model 3 adds in the two secondary variables of interest, ENC and PNC. From model 1 we found that components of ENC, child PNC and pregnancy type were statistically significant predictors of ENNM without adjusting for other covariates, but place of delivery was not significantly associated with ENNM.

Adjusting for neonatal, pregnancy care, socio-demographic, and community characteristics did not affect the relationship of delivery care with ENNM, which remained statistically not significant (model 2). Pregnancy type and birth order were significantly associated with ENNM.

In model 3, we found that the newborns who received high ENC (i.e., at least 3 ENC components) had 56% lower odds of early neonatal death (aOR: 0.44; 95% CI: 0.24–0.81) compared to those who received low ENC (i.e., 0–2 components). The odds of dying in the early neonatal period are 68% lower when the newborn received PNC after birth (aOR: 0.32; 95% CI: 0.16–0.64) compared to their counterparts who did not receive any PNC. Singleton births have 95% lower odds of ENNM than the twin births (aOR: 0.05; 95% CI: 0.01–0.16). In this adjusted model, neonates delivered at a health facility had 49% higher odds of ENNM (aOR: 1.49; 95% CI: 0.77–2.89), but this result is not statistically significant. Other findings are similar to model 2 except that the adjusted odds ratio for the birth order '2–4' was marginally significant at the conventional 5% level ($p = 0.051$).

## Discussion

Using data on 4,624 live births in Bangladesh we found 69 deaths within the first week of life, yielding an ENNM rate of 15.7 per 1,000 live births among infants delivered at a health facility and 14.4 per 1,000 live births among those delivered at home. These deaths were most concentrated in the lowest wealth quintile, a population that was also less likely to experience a facility

birth. The first 24 hours after delivery are crucial to both newborns and mothers–this study finds that 46% of neonatal deaths occurred during this period. Our analysis identified several factors that are associated with ENNM; however, delivering in a health facility was not significantly associated with the risk of early neonatal death in any of the models.

Selection effects whereby women who experience high-risk pregnancies or complications during delivery may be more likely to deliver in a health facility likely contribute to ENNM among facility deliveries, which could attenuate potential protective effects of facility delivery. We are not able to control for differences in such unobserved confounding obstetric risk factors in our analysis. However, recently published results from the 2017–18 BDHS shows that the percentage of births delivered in a health facility increased from 37% in 2014 to 50% in 2017–18, yet the neonatal mortality rate increased from 28 deaths per 1,000 live births in 2014 to 30 deaths per 1,000 live births in 2017–18 [32]. This indicates that, at the population level, higher levels of facility delivery do not necessarily translate into increased survival of newborns, which is consistent with our finding of no association between facility delivery and newborn survival.

The quality of delivery and newborn care is an important factor that could contribute to our null findings and to these population-level trends [17, 33]. In our analysis, 24% of newborns who were delivered in a health facility received low ENC (0–2 components of ENC). According to the 2014 Bangladesh Health Facility Survey (BHFS), only 17% of facilities that provide delivery services performed all seven signal functions for basic emergency obstetric and neonatal care (BEmONC), and only 6% performed all nine signal functions for comprehensive obstetric and neonatal care (CEmONC) [34]. The overall preparedness of facilities to provide high-quality vaginal delivery care was very low—only 3% of health facilities demonstrated service-specific readiness in terms of WHO-recommended minimum requirements for quality services [35]. Lack of trained staff, guidelines, life-saving medicines and partographs was more pronounced than the lack of other essential items for providing quality normal delivery services [34]. Facility deliveries will not save newborn lives if health facilities are not able to provide life-saving care in an emergency. This phenomenon has also been seen in other South Asian countries [36].

The existing evidence indicates that ENC practices can contribute to reducing early-neonatal morbidity and mortality in Bangladesh [15, 37, 38]. Our analysis also found that a higher reported level of ENC is significantly associated with increased early newborn survival. A review of recent research found that a significant proportion of neonates born at home in low-and-middle-income countries experienced hypothermia, which contributed as a comorbidity to the major causes of neonatal deaths such as infection, asphyxia, and preterm birth [17]. ENC can address these causes of early neonatal mortality in infants. In our study 30% of all newborns received less than three components of ENC, and this was higher among infants born at home (34%) compared to infants born in a health facility (24%). Recent study findings have shown that women can not necessarily recall the exact timing or sequence of events (e.g., components of ENC; PNC etc.) during or immediately after delivery, leading to recall bias in newborn care indicators, which needs to be kept in mind when interpreting these results [39]. Nevertheless, our study provides evidence that there is scope to improve the quality of ENC among both home and facility births and that such improvement could reduce ENNM in Bangladesh.

PNC provides an opportunity to assess and treat complications in newborns and to counsel mothers on how to care for themselves and their newborns. We found that infants who reportedly received any PNC experienced significantly lower early neonatal mortality than those who did not, consistent with other studies in Bangladesh and neighboring countries [38, 40, 41]. One caveat in interpreting these results, however, is that those infants who die

immediately around the time of delivery will not survive long enough to receive PNC. We reran the models excluding infants who died in the first 24 hours after delivery and found PNC was still highly significant. Most PNC first occurs within the first 48 hours after the delivery when it is reported. We found that among all live births, 48% of infants born at home and 18% of infants born in health facilities were reported to have not received any PNC; increasing access to high-quality early PNC remains an important intervention to reduce ENNM.

Although not the focus of our study, we found that ENNM is substantially lower among singleton births compared to twin births, as expected. Previous studies suggest that twin pregnancies are associated with an eight to ten-fold increase in the perinatal mortality rate, mostly due to preterm birth [39] and its associated complications [42–44]. Intensive antenatal and delivery care are essential for twin pregnancies, but we found that 46% of the twins in our sample were delivered at home suggesting a need to increase delivery care and associated ENC and PNC among multiple births.

## Strengths and limitations

This study has several limitations, some of which have already been touched on above. The reliability of mortality estimates calculated from retrospective birth histories depends upon the extent to which birth dates and ages at death are accurately reported and recorded. Omission of either births or deaths is a serious problem since they affect the accuracy of the mortality estimates. Errors in reporting birth dates may cause a distortion of trends over time, while errors in reporting of age at death can distort the age pattern of mortality. Omission of early neonatal deaths is of particular concern in DHS birth history data [7] and could lead to underestimation of ENNM in this study [7, 45]. Second, as noted above, newborns must survive long enough to receive PNC so some instances of lack of newborn care may be due to early neonatal death rather than the other way round. Third, there is known recall bias in women's reports of both maternal and newborn care at the time of delivery [46]. Fourth, our results are subject to unmeasured/residual confounding by factors not measured by the DHS. Fifth, though BDHS 2014 collects data from a representative sample of ever married women and subsequent child births, given the rarity of the event we only got 69 early neonatal deaths in our study sample. Finally, information on pregnancy and delivery complications were not captured in the BDHS 2014 so we are unable to control for obstetric risk factors that could affect both ENNM and the choice to deliver in a health facility. For all these reasons, the study results should be interpreted as indicating an association between the delivery and newborn care and ENNM, not causation.

Despite these limitations, this study has many strengths. The BDHS data are nationally representative and have a high (98%) response rate [7]; as a result, sample selection bias is unlikely to affect the study findings. The BDHS used a well-tested, standardized questionnaire and employed a high-quality data collection process to reduce interview bias and to identify and address field data collection problems rapidly.

## Conclusion

This study provides evidence on the association between ENNM and place of delivery and newborn care at the population level for national policy makers and health experts who aim to reduce ENNM in Bangladesh. Our study findings indicate that increasing facility delivery is not sufficient to reduce ENNM by itself. Comprehensive ENC and PNC are significantly associated with early newborn survival. Standard operating procedures (SOPs) to ensure the quality of care during delivery, to ensure all components of ENC, and to provide high-quality early PNC are of critical importance to realize the theoretical potential of facility delivery to avert

neonatal deaths. Further studies are required to identify the causes of ENNM and relevant risk factors.

## Supporting information

**S1 Fig.**
(TIF)

## Acknowledgments

The authors would like to thank the DHS Program for granting access to BDHS datasets and making publicly available the published DHS reports. The views expressed herein are solely those of the authors and do not necessarily reflect the views of any institution or organization.

## Author Contributions

**Conceptualization:** Rashida-E Ijdi, Katherine Tumlinson, Siân L. Curtis.

**Data curation:** Rashida-E Ijdi.

**Formal analysis:** Rashida-E Ijdi, Siân L. Curtis.

**Methodology:** Rashida-E Ijdi, Katherine Tumlinson, Siân L. Curtis.

**Supervision:** Katherine Tumlinson, Siân L. Curtis.

**Validation:** Siân L. Curtis.

**Visualization:** Rashida-E Ijdi.

**Writing – original draft:** Rashida-E Ijdi.

**Writing – review & editing:** Katherine Tumlinson, Siân L. Curtis.

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
