## [Decision Letter · Decision Letter 0]

14 Jan 2021

PONE-D-20-30510

The role of institutional delivery and newborn care in preventing early-neonatal mortality in Bangladesh

PLOS ONE

Dear Dr. Ijdi,

Thank you for submitting your manuscript to PLOS ONE. After careful consideration, we feel that it has merit but does not fully meet PLOS ONE’s publication criteria as it currently stands. Therefore, we invite you to submit a revised version of the manuscript that addresses the points raised during the review process.

We look forward to receiving your revised manuscript.

Kind regards,

Calistus Wilunda, DrPH

Academic Editor

PLOS ONE

Additional Editor Comments:

In the abstract (and maybe elsewhere) it is not explicit who are “their counterparts”.

Use multivariable instead of multivariate. To distinguish between these two terms, please refer this this https://academic.oup.com/ntr/advance-article/doi/10.1093/ntr/ntaa055/5812038

In descriptive results in the tables, clarify whether the frequencies and percentages are weighted or unweighted

It is unclear whether all the survey design features, including weighting and stratification, were taken into account.

P values of 0.000 should be written as <0.001

In Table 3, please present the results of the pre-specified exposure variables first.

In presenting (and discussing) the results, please focus on the exposure variables of interest. This is because the multivariable model includes some variables that are on the causal pathway. For example, size of child at birth, a proxy for birthweight, is most likely a mediator between maternal socio-demographic characteristics and ENNM. Thus, the ORs for the socio-demographic characteristics may be biased.

Journal Requirements:

2. In statistical methods, please clarify whether you corrected for multiple comparisons.

3. In your statistical analyses, please state whether you accounted for clustering by region/ state. For example, did you consider using multilevel models? If not, please provide your rationale for not doing so.

5. We note that Figure 1 in your submission contain map images which may be copyrighted. All PLOS content is published under the Creative Commons Attribution License (CC BY 4.0), which means that the manuscript, images, and Supporting Information files will be freely available online, and any third party is permitted to access, download, copy, distribute, and use these materials in any way, even commercially, with proper attribution. For these reasons, we cannot publish previously copyrighted maps or satellite images created using proprietary data, such as Google software (Google Maps, Street View, and Earth). For more information, see our copyright guidelines: http://journals.plos.org/plosone/s/licenses-and-copyright.

5.1.    You may seek permission from the original copyright holder of Figure 1 to publish the content specifically under the CC BY 4.0 license. 

5.2.    If you are unable to obtain permission from the original copyright holder to publish these figures under the CC BY 4.0 license or if the copyright holder’s requirements are incompatible with the CC BY 4.0 license, please either i) remove the figure or ii) supply a replacement figure that complies with the CC BY 4.0 license. Please check copyright information on all replacement figures and update the figure caption with source information. If applicable, please specify in the figure caption text when a figure is similar but not identical to the original image and is therefore for illustrative purposes only.

Reviewers' comments:

Reviewer's Responses to Questions

**Comments to the Author**

1. Is the manuscript technically sound, and do the data support the conclusions?

Reviewer #1: Yes

Reviewer #2: No

Reviewer #3: Yes

2. Has the statistical analysis been performed appropriately and rigorously? 

Reviewer #1: Yes

Reviewer #2: No

Reviewer #3: Yes

3. Have the authors made all data underlying the findings in their manuscript fully available?

Reviewer #1: Yes

Reviewer #2: Yes

Reviewer #3: Yes

4. Is the manuscript presented in an intelligible fashion and written in standard English?

Reviewer #1: Yes

Reviewer #2: Yes

Reviewer #3: Yes

5. Review Comments to the Author

Reviewer #1: Comments are given in the manuscript

Comments are given in the manuscript. Comments are given in the manuscriptComments are given in the manuscriptComments are given in the manuscriptComments are given in the manuscriptComments are given in the manuscript

Reviewer #2: PONE-D-20-30510

The role of institutional delivery and newborn care in preventing early-neonatal mortality in Bangladesh

The study has several strengths and significant implications, but these were not highlighted in the manuscript. There are many key issues. Please see below for details.

Major comments

1. The abstract reflects the title, but not the manuscript. The authors could structure the article based on the title and the abstract. Exploring a single exposure-outcome relationship could be more interesting than exploring (many) factors associated with the outcome.

2. The rationale of the study idea is vague. The authors claimed that only a few studies had been done, and most of them are small-scale and clinically oriented. What is the problem of clinically-oriented studies? Ref 17 used BDHS 2011, Ref 19 used 15 times larger sample size than this study, Ref 26 had results from cluster randomized controlled trials, and Ref 27 used BDHS 2004, 2007, and 2011 (and they had approximately four times larger sample size than this study).

3. The study objective is ambiguous. The title says the authors explored the relationship of institutional delivery and newborn care with early neonatal mortality in Bangladesh. But Page 5 (line 4) says differently. The authors even confuse the readers on Page 5 (line 7) and Page 6 (line 4-5). This confusion remained a significant issue throughout the manuscript. However, having a single objective, e.g., exploring the relationship between institutional delivery and early neonatal mortality, could be more interesting.

4. The introduction is unnecessarily big. This section could have only a maximum of 10–15% of the manuscript’s total word count (https://doi.org/10.1016/j.jclinepi.2013.01.004).

5. To provide the scientific community with novel results, the authors could redo the study with more recent survey data (e.g., BDHS 2017-18).

6. The conceptual framework is incomplete. The authors missed many relevant covariates, e.g., maternal BMI, occupation, birth order, birth interval, previous death of any siblings, number of ANC, type of delivery, paternal education and occupation, etc. (https://doi.org/10.1371/journal.pone.0221503;
https://bmjopen.bmj.com/content/5/8/e006722;
https://archpublichealth.biomedcentral.com/articles/10.1186/s13690-017-0224-6). Please include all relevant covariates or use stepwise with AIC in the model building. However, the better option is to use a directed acyclic graph to identify potential confounders and risk factors (https://www.nature.com/articles/s41390-018-0071-3;
https://link.springer.com/article/10.1007%2Fs10654-019-00494-6).

7. Page 6, line 4-12: The hypothesis part should be in the Introduction section (just after the study aim) instead of the Methods.

8. The methods used are technically incorrect. The authors must use the design-based Rao-Scott test instead of Pearson’s Chi-square test to account for the complex survey design. Also, the authors only considered sampling weights in the regression analysis. But they did not consider all the complex survey design-features (e.g., strata, cluster, sampling weights).

9. Model diagnostics are missing. Also, model validation or sensitivity analysis is missing. The authors could use the generalized estimating equation (GEE) or propensity score matching/weighting as a sensitivity analysis for the main findings.

10. The analysis strategies are mostly not understandable and reproducible. Also, all the methods were not introduced or at least mentioned/cited in the Methods section before discussing them in the Results/Discussion section.

11. Tables should be stand-alone so that readers can understand a table without reading the text and vice versa.

12. The results interpretation both in the Result section and Discussion section is somewhat poor. There is too much description of confounded results (Table 1 to 3). Page 10 (line 8-17): wrong interpretation. Trend analysis is not possible from this study, but the authors interpreted it as “increases/decreases.” This should be “higher/lower.”

13. Table 4 is very interesting. But the interpretation of the results is not adequate. There is Table 2 fallacy (https://doi.org/10.1093/aje/kws412).

14. The first paragraph of the Discussion section does not match with the Result section. Overall, the logical integration of concepts (from introduction to discussion) needs to be improved. The authors should concisely write the paper. There are many irrelevant discussions throughout the manuscript. In contrast, the study implications and future directions are missing. The rate of the quality of contribution in the literature and strong implications are not adequate. Also, the study strength is not well articulated.

Minor changes

15. The outcome is extremely rare (the prevalence is ~1.4%). The authors could aggregate two/three cycles to improve statistical power.

16. Need references in categorizing many of the covariates, particularly when constructing them. Otherwise, please provide validation of these constructions.

17. Page 2, line 2-3: Paraphrase the line. The line is just a copy from the cited article.

18. Figure 1: Not sure why needed. The authors could drop this figure.

19. Page 3, line 9-11:” Update the statistics based on BDHS 2017-18.

20. Multicollinearity is not a major problem in causal models. Instead, we want to reduce confounding bias regardless of collinear variables.

21. Table 1: Some % do not add up to 100%.

22. Tables and figures could be improved. The figure's quality is too low, and the tables are not of publication quality. Please drop vertical lines and put only 3/4 horizontal lines based on the objective for each table.

23. Page 13, line 4: The odds of ENNM was…

24. Page 14, line 2: … significantly associated with ENN (or significantly associated with lower odds of ENNM).

25. Put total/overall just after the heading of the table, not at the bottom.

26. Drop stars from the tables. The authors have used p-values in three decimal points. Please use <0.001 if any p-value is less than 0.001.

27. Add a limitation: The temporality of the association cannot be made due to the nature of the study design.

Reviewer #3: The manuscript deals with identifying the factors affecting early neonatal mortality (ENNM), which represents more than 80\\% of all neonatal mortality in Bangladesh, with a specific focus on place of delivery and newborn care. The 2014 Bangladesh Demographic and Health Survey data were used for assessing early-neonatal survival in children born in the three years preceding the survey. Multivariate logistic regression models have been used. The study findings highlight the importance of newborn and postnatal care in preventing early neonatal deaths.

I suggest some revisions. Following questions must be needed to answer.

1. Any variables about husband have not been used. Particularly, husband's education, or parental education, employment status, occupation are important to consider. Besides, mother's age at first birth is important.

2.Why 2018 BDHS data has not been used for this study instead of BDHS 2014 data?

3. How about the death rate among the missing cases? Total 309 cases have been omitted. Just in case if considerable amount of death rate is found among 309 cases, then suitable imputation method could be used to impute the missing observations.

4. Have the twin babies considered here. If so then, whether it could be an important factor in the model.

5. In Statistical Analysis section, Variance inflation factors (VIF) were assessed to examine collinearity between variables before entering them into the multivariable models? What cut off point was used here?

6. The confounding could be existed here. Particularly, ANC visits and ENNM might be under important consideration in making such unobserved confounding. What is the response of authors in this regards?

7. Is simple map as displayed in Figure 1 important? Rather authors could have mapping for Bangladesh by disagegating to division levels. Authors could use many software including ``mapReasy" R package (Islam et al., 2017).

6. PLOS authors have the option to publish the peer review history of their article (what does this mean?). If published, this will include your full peer review and any attached files.

Reviewer #1: No

Reviewer #2: **Yes: **Md. Belal Hossain

Reviewer #3: **Yes: **Md Hasinur Rahaman Khan

---

## [Author Response · Author response to Decision Letter 0]

3 Jun 2021

Reviewers' comments:

Reviewer #1: 

1. Page 1, line 13: So the issue is that if a mother receive higher level of essentials newborn care component (HLENCC) then they are less likely to die”. 

1. Here higher level of essential newborn care components are not defined

2. availing higher level of ENCCs are not possible for many mother (this is problem of demand side due to low socio-economic and other factors);

3. providing higher level of ENCCs are not possible by the providers (this is service side problems/limitations);

4 so, even if you suggest a line for policy makers, it is not possible for policy makers to execute 

Response: In this study, we have divided the essential newborn care (ENC) components into two categories: low-level of ENC (i.e., receiving two or fewer components of ENC) and high-level of ENC (i.e., receiving three or more ENC components). The different components of ENC include the use of sterilized instruments to cut the umbilical cord, applying nothing to the cord, immediate drying (within 5 minutes) keeping the baby warm, delaying bathing to 72 hours after birth, and initiating breastfeeding within 1 hour of delivery. Each component was weighted equally, and a score was developed for the number of ENC components received by the infant (0-5). We described this process in detail on p7 lines 16-21; p8 lines 1-2. 

We fully agree with the reviewer that specific groups of women – especially those who are low income – may experience barriers to ENC and other facility-level services. We, however, believe that the service-side problems can be addressed by better planning and systems improvement (e.g., HR, supply-chain management, etc.). For this reason, the policy recommendations throughout our discussion section are intended to encourage policymakers to increase access to these services for all women. We have added text to emphasize access for low income populations in the abstract and discussion section. 

2. Page 1, line 16 : Same as above

Response: We have specified the counterpart as ‘who had not received any PNC’ in p1, lines 18-20. 

3. Page 1, line 21: Non specific, please specify what you meant by Accelerated improvement? What a policy maker would understand by this?

Response: We would like to thank the reviewer for this comment. We have revised the statement in p2, line 4-5. as ‘Sustained efforts to expand access to high quality essential newborn and postnatal care are needed in health facilities, particularly in facilities serving low-income populations.’ 

4. Conceptual framework, data analysis are consistent 

Response: We would like to thank the reviewer for this comment. 

5. In Table 2 Can you reanalyze the deaths by rural and urban and see where the deaths are higher??

Response: Table 1 shows that 53 (77.3%) of early neonatal deaths is in rural areas, and 16 (22.7%) of the early neonatal deaths are in urban areas. The numbers of early neonatal deaths is too small to permit the analysis in Table 2 to be presented by urban and rural residence.

6. Early Neonatal Death is high at home delivery but early neonatal deaths on days 1-6 after birth is surprisingly lower at home delivery. How to explain

Response: One plausible explanation is that more complicated cases might be seeking care at facilities. We have included this explanation in our discussion section on p18, lines 7-16; p18, lines 17-21; p19, lines 1-8. 

A related plausible explanation is that neonates born with complications could survive longer in a facility setting due to treatment but still die, shifting the death from the first day to 1-6 days.

7. In Table 3 Sample from urban is much lower that rural.

Response: The sample in the BDHS is selected to be representative of the population of Bangladesh. The majority of the people of Bangladesh resides in rural areas and fertility is higher in rural areas. Hence, the rural sample is notably larger than the urban sample. 

Reviewer #2: 

The role of institutional delivery and newborn care in preventing early-neonatal mortality in Bangladesh

The study has several strengths and significant implications, but these were not highlighted in the manuscript. There are many key issues. Please see below for details.

Thank you for your careful reading and generous comments on our paper.

Major comments

1. The abstract reflects the title, but not the manuscript. The authors could structure the article based on the title and the abstract. Exploring a single exposure-outcome relationship could be more interesting than exploring (many) factors associated with the outcome.

Response: We thank the reviewer for the comment. We have restructured the abstract and manuscript to better reflect the title and to better represent the information we intend to convey with this analysis. 

2. The rationale of the study idea is vague. The authors claimed that only a few studies had been done, and most of them are small-scale and clinically oriented. What is the problem of clinically-oriented studies? Ref 17 used BDHS 2011, Ref 19 used 15 times larger sample size than this study, Ref 26 had results from cluster randomized controlled trials, and Ref 27 used BDHS 2004, 2007, and 2011 (and they had approximately four times larger sample size than this study).

Response: We thank the reviewer for highlighting this point. For this study we have selected BDHS 2014 as a nationally representative sample that could be generalizable to the population of Bangladesh. In general, clinically-oriented studies, while valuable, are limited in their generalizability. Our use of the BDHS 2014 data addresses the following limitations of prior studies: 

- Ref 17 (now 15) focused on essential newborn care only by delivery locations using BDHS 2011. 

- Ref 19 (now 17) used data from a study conducted in the rural areas of Matlab sub-district, covering about 250,000 people. 

- Ref 26 (now 24) results from community-based surveillance of perinatal events and verbal autopsies in 18 rural inions of Bogra, Moulavibazar, and Faridpur (out of 4,562 unions in Bangladesh). Neither of the studies is nationally representative nor cover urban areas. 

- Ref 27 (now 25) used three rounds of BDHS (2004, 2007, 2011) with the focus on effects on child mortality of different combinations of taking iron-folic acid supplementation, tetanus toxoid vaccination and antenatal care visits during pregnancy. 

In our study we have focused on the association between place of delivery and early neonatal mortality, where essential newborn care and postnatal care are on the causal pathways as the secondary exposures. Addressing this relationship in Bangladesh in a nationally representative dataset is a unique contribution of our analysis that complements and expands on these other studies. We emphasized this in the discussion section. 

3. The study objective is ambiguous. The title says the authors explored the relationship of institutional delivery and newborn care with early neonatal mortality in Bangladesh. But Page 5 (line 4) says differently. The authors even confuse the readers on Page 5 (line 7) and Page 6 (line 4-5). This confusion remained a significant issue throughout the manuscript. However, having a single objective, e.g., exploring the relationship between institutional delivery and early neonatal mortality, could be more interesting.

Response: We have rephrased the purpose of the study to make it clear on p. 5 (lines 8-9): “This study aims to explore the association of the place of delivery and newborn care with ENNM in Bangladesh.” We have reorganized the paper to focus more clearly on this objective, as described further below.

4. The introduction is unnecessarily big. This section could have only a maximum of 10–15% of the manuscript’s total word count (https://doi.org/10.1016/j.jclinepi.2013.01.004).

Response: Thank you for this comment. We have substantially reduced the length of the introduction (from 4 pages to 2.5 pages) and it is now approximately 12% of the manuscript’s total word count. 

5. To provide the scientific community with novel results, the authors could redo the study with more recent survey data (e.g., BDHS 2017-18).

Response: We thank the reviewer for this comment. After careful consideration we concluded that redoing the study with recent BDHS 2017-18 data would be a great follow on for this paper. We think that this analysis of the BDHS 2014 data provides a valuable contribution to the literature on delivery and newborn care and ENNM in Bangladesh in the MDG period and provides a foundation to build on with further analysis of the BDHS 2017-18 data (which was released after we submitted this paper) to explore whether and how these relationships have changed as facility delivery has increased in Bangladesh. Therefore, we will explore replicating the analysis with the BDHS 2017-18 data in a separate paper.

6. The conceptual framework is incomplete. The authors missed many relevant covariates, e.g., maternal BMI, occupation, birth order, birth interval, previous death of any siblings, number of ANC, type of delivery, paternal education and occupation, etc. (https://doi.org/10.1371/journal.pone.0221503;
https://bmjopen.bmj.com/content/5/8/e006722;
https://archpublichealth.biomedcentral.com/articles/10.1186/s13690-017-0224-6). Please include all relevant covariates or use stepwise with AIC in the model building. However, the better option is to use a directed acyclic graph to identify potential confounders and risk factors (https://www.nature.com/articles/s41390-018-0071-3;
https://link.springer.com/article/10.1007%2Fs10654-019-00494-6).

Response: We thank the reviewer for this comment. After careful review we have added paternal education and twin births in our model based on the directed acyclic graph (DAG) that we have added as an appendix. We agree that other variables mentioned by the reviewer have been shown to be associated with neonatal and infant survival but they are not necessarily also strongly associated with delivery care to be confounders of the relationship between delivery care and early neonatal mortality. Based on best practice, our preference is to include only the minimally sufficient adjustment set and avoid overfitting the model. In addition, previous birth interval and survival of the preceding child are not defined for first births and our sample includes first births. We found collinearity between birth order & birth interval; ANC by medically trained providers & number of ANC; and place of delivery & type of delivery. Therefore, we have not included birth interval, number of ANC and the type of delivery.

7. Page 6, line 4-12: The hypothesis part should be in the Introduction section (just after the study aim) instead of the Methods.

Response: The hypothesis part has been moved as appropriate in p7, line 12-13. 

8. The methods used are technically incorrect. The authors must use the design-based Rao-Scott test instead of Pearson’s Chi-square test to account for the complex survey design. Also, the authors only considered sampling weights in the regression analysis. But they did not consider all the complex survey design-features (e.g., strata, cluster, sampling weights).

Response: We thank the reviewer for this suggestion. As described in our methods section, we have used Rao-Scott test instead of Pearson’s Chi-square test to account for the complex survey design. We have clarified that the appropriate sampling weights were applied to all analyses using Stata’s survey estimation procedure (svy) to achieve nationally representative estimates of the population after adjusting for sample strata and clusters, p8, lines 20-21; p9, lines 1-18. 

9. Model diagnostics are missing. Also, model validation or sensitivity analysis is missing. The authors could use the generalized estimating equation (GEE) or propensity score matching/weighting as a sensitivity analysis for the main findings. 

Response: Thank you for this comment. Regarding model diagnostics, we agree with the reviewer that it is important to test the assumption that our independent variables are not highly correlated with each other, as correlation between independent variables could be a potentially significant issue. Given the large number of covariates, multicollinearity was assessed using variance inflation factors (VIFs). All covariates were acceptable with VIFs less than 5. Since correlations between variables were not a concern, all covariates were retained in the final model. These steps are clarified in the revised manuscript at the end of the Methods section. As further evidence that multicollinearity is unlikely, the model parameter estimates (point and standard errors) appear stable. 

Regarding generalized estimating equation (GEE), we agree that using GEE provides a benefit in terms of adjusting for clustering. However, in this analysis we have used the standard survey procedure for DHS data, which goes beyond accounting for clustering and incorporates the two-stage sampling strategy. Data for our analysis come from a cross-sectional survey, where the DHS samples are stratified by geographic region and by urban/rural areas within each region. Within each stratum, the sample is designed and selected independently. We had no reason to suspect the independence of observations in this study, and a key assumption underpinning GEE is that the independence of observations does not hold (e.g., using longitudinal/repeated measures analysis). Failure to take into account correlation between observations would lead to the regression estimates being less efficient. We also accounted for stratification and clustering in our analysis and do not think GEE would be provide any added benefit.

Regarding propensity score matching/weighting, we agree that this can be a useful way to adjust for measured confounding. However, we have used multivariable logistic regression as the adjustment method in this analysis. For clarity on our adjustment technique, we have now added text to indicate that, in specifying our model, we used a DAG to identify potential confounders and select a minimally sufficient adjustment set. (added citation: Greenland S, Glymour M. Causal Diagrams. Modern Epidemiology. 3rd ed. Philadelphia: Lippincott Williams & Wilkins; 2008. p. 183–209.) We have also added a sentence to the limitations section to clarify that our results are subject to unmeasured/residual confounding by factors not measured by the DHS.

To our understanding, model validation is mainly used for prediction models and is therefore not relevant for this analysis. 

10. The analysis strategies are mostly not understandable and reproducible. Also, all the methods were not introduced or at least mentioned/cited in the Methods section before discussing them in the Results/Discussion section.

Response: Thank you for this comment. We have made substantial changes throughout the methods, results, and discussion section to clarify our approach, and to ensure that our methods, results, and discussion follow a logical flow. Please see responses to prior comments for additional details of changes made based on this reviewer’s helpful advice. 

11. Tables should be stand-alone so that readers can understand a table without reading the text and vice versa.

Response: We have updated the titles of Tables 1, 3a (formerly Table 3) and 4 to ensure readers can understand the tables without reading the text. We have followed PLOS One’s guidelines for creating tables. 

12. The results interpretation both in the Result section and Discussion section is somewhat poor. There is too much description of confounded results (Table 1 to 3). Page 10 (line 8-17): wrong interpretation. Trend analysis is not possible from this study, but the authors interpreted it as “increases/decreases.” This should be “higher/lower.”

Response: We have substantially revised the results interpretation in the Results section by cutting out the discussion of confounded results in Table 3a, and 4 and focused on the results for our primary and secondary variables of interest (delivery care, ENC and PNC). We agree that trend analysis is not possible from this study and have generally used the terms high/lower but have retained the increased/decreased language in some contexts when referring to relationships with ordinal variables where the increase/decrease is related to the increase/decrease in the ordinal variable not increase/decrease over time (e.g. the percentage of women who delivered in a health facility increases as the number of ANC components received increases (p12 lines 16-18)).

13. Table 4 is very interesting. But the interpretation of the results is not adequate. There is Table 2 fallacy (https://doi.org/10.1093/aje/kws412).

Response: We would like to thank the reviewer for this comment. We have revised the discussion of Table 4 to focus on the three exposure variables of interest (delivery care, ENC and PNC), p15, lines 3-9, p17, lines 1-14. Please also see our response to the previous comment.

14. The first paragraph of the Discussion section does not match with the Result section. Overall, the logical integration of concepts (from introduction to discussion) needs to be improved. The authors should concisely write the paper. There are many irrelevant discussions throughout the manuscript. In contrast, the study implications and future directions are missing. The rate of the quality of contribution in the literature and strong implications are not adequate. Also, the study strength is not well articulated.

Response: The first paragraph of the Discussion aligns with Table 2 in the Results. We have substantially revised the Introduction, Methods and Results sections to tighten the logical flow of the paper, as described in our responses to previous comments. In particular, we have clarified the objectives of the study, added a DAG as an appendix and focused the discussion of the results on the three main exposure variables of interest: delivery care, ENC and PNC. We have also made some adjustments to the analysis to strengthen the integration of the introduction, methods, and results. Specifically, we have divided Table 3 into two tables to align with the DAG; Table 3a examines the bivariate relationships with the confounders in the upper part of the DAG while Table 3b examines the association between delivery care and ENC and PNC in the lower part of the DAG. Model 2 in Table 4 has also been updated to include delivery care plus the confounders in the upper part of the DAG and then Model 4 adds ENC and PNC to that model following the lower part of the DAG. We have also worked to ensure that the language used in presenting and describing the results is consistent with language used when describing the conceptual model. 

In the conclusion we have highlighted the main implications of the study: “Our study findings indicate that increasing facility delivery is not sufficient to reduce ENNM by itself. Comprehensive ENC and PNC are significantly associated with early newborn survival. Standard operating procedures (SOPs) to ensure the quality of care during delivery, to ensure all components of ENC, and to provide high-quality early PNC are of critical importance to realize the theoretical potential of facility delivery to avert neonatal deaths.” (p.22 line 4–9). We are reluctant to make stronger policy recommendations because there are a number of limitations with the study that limit the casual interpretation of the results, as described in the Discussion (p.20 line 20-21; p.21 lines 1- 14).

Minor changes

15. The outcome is extremely rare (the prevalence is ~1.4%). The authors could aggregate two/three cycles to improve statistical power.

Response: We thank the reviewer for this comment. We decided against aggregating two/three rounds of BDHS data to increase the power because the reference period would be very wide (8-10 years), limiting the usefulness of results. In addition, the proportion of women delivering in health facilities has been increasing over time so unobserved selection effects could be changing over time which could affect the relationship between delivery care and early neonatal mortality over time. This itself would be interesting to explore but we think that would be a different paper that would build on this one. 

16. Need references in categorizing many of the covariates, particularly when constructing them. Otherwise, please provide validation of these constructions.

Response: Most of the categorical variables follow categories that are widely used in the literature and reflect well-established relationships (sex of child, pregnancy type, ANC with MTP, mother’s age, mother’s education, paternal education, wealth quintile, place of residence, distance to facility). The two constructed variables are number of components of ANC received and number of components of ENC received. The number of components of ANC and ENC follows components of care reported by DHS, which is informed by international standards for these indicators (Ref: https://www.healthynewbornnetwork.org/hnn-content/uploads/Newborn-Care-Indicators-for-Household-Surveys_HNN_10Jan2013.pdf ). For ANC care, we created categories based roughly on quartiles. This approach also allows us to distinguish those who received the extremes of no components vs all components. For ENC we distinguish low care from higher care. The decision to divide this as 0-2 vs 3+ components was based on the distribution of the indicators such that we were able to distinguish very low care while also maintaining a reasonably large sample size in the lower category.

We set our categories of the covariates, in the Data Analysis section, in such a way to evenly split the whole sample, p7, lines 16-21; p8, lines 1-2.

17. Page 2, line 2-3: Paraphrase the line. The line is just a copy from the cited article.

Response: This sentence has now been edited as ‘Early neonatal mortality (ENNM), defined as the death of a newborn between zero and six days after birth, represents 73% of all neonatal deaths (i.e., deaths occurring during 0-27 days of life) worldwide [1].’ 

18. Figure 1: Not sure why needed. The authors could drop this figure.

Response: We have dropped Figure 1.

19. Page 3, line 9-11:” Update the statistics based on BDHS 2017-18.

Response: After careful review we have decided to drop this line to reduce the length of introduction.

20. Multicollinearity is not a major problem in causal models. Instead, we want to reduce confounding bias regardless of collinear variables.

Response: Thank you for this comment. We agree that controlling for confounding is more important in causal modeling than multicollinearity. We have added a DAG in an annex to the paper and strengthened the description of the linkages between the conceptual model, DAG and analysis while aiming for a minimally sufficient adjustment set of confounders, as described in our responses to previous comments.

21. Table 1: Some % do not add up to 100%.

Response: We have checked the table and the percentages now sum to 100%.

22. Tables and figures could be improved. The figure's quality is too low, and the tables are not of publication quality. Please drop vertical lines and put only 3/4 horizontal lines based on the objective for each table.

Response: We have followed PLOS One’s guidelines for formatting tables and dropped Figure 1. 

23. Page 13, line 4: The odds of ENNM was…

Response: We have made this correction in p17, lines 5-14.

24. Page 14, line 2: … significantly associated with ENN (or significantly associated with lower odds of ENNM).

Response: We would like to thank the author for pointing this out. We have done the correction as ‘In model 3, we found that the newborns who received high ENC (i.e. at least 3 ENC components) had 56% lower odds of early neonatal death (aOR: 0.44; 95% CI: 0.24-0.81) compared to those who received low ENC (i.e., 0-2 components)’ in p17, lines 5-7.

25. Put total/overall just after the heading of the table, not at the bottom.

Response: We have moved total/overall just after the heading of the table.

26. Drop stars from the tables. The authors have used p-values in three decimal points. Please use <0.001 if any p-value is less than 0.001.

Response: We have made these changes in the relevant tables (in Tables 3a, 3b and 4).

27. Add a limitation: The temporality of the association cannot be made due to the nature of the study design.

Response: By definition, delivery care precedes early neonatal death temporally even in this cross-sectional study but complications during pregnancy and delivery can lead to a decision to deliver in a facility and may also be associated with increased risk of early neonatal death. We have noted this limitation on p. 21, lines 10-14: “Finally, information on pregnancy and delivery complications were not captured in the BDHS 2014 so we are unable to control for obstetric risk factors that could affect both ENNM and the choice to deliver in a health facility”. In addition, lack of PNC and ENC can be the result of death immediately after delivery so the temporality of the association between ENC and PNC and early neonatal death cannot be determined in this study. We have noted this limitation in the Discussion on p.20 lines 5-6 : “One caveat in interpreting these results, however, is that those infants who die immediately around the time of delivery will not survive long enough to receive PNC” and on p. 21 lines 6-8: “Second, as noted above, newborns must survive long enough to receive PNC so some instances of lack of newborn care may be due to early neonatal death rather than the other way round.”

Reviewer #3: 

The manuscript deals with identifying the factors affecting early neonatal mortality (ENNM), which represents more than 80% of all neonatal mortality in Bangladesh, with a specific focus on place of delivery and newborn care. The 2014 Bangladesh Demographic and Health Survey data were used for assessing early-neonatal survival in children born in the three years preceding the survey. Multivariate logistic regression models have been used. The study findings highlight the importance of newborn and postnatal care in preventing early neonatal deaths.

I suggest some revisions. Following questions must be needed to answer.

We thank the reviewer for the review and thoughtful comments. on our paper. The responses to each of your comments are given below:

1. Any variables about husband have not been used. Particularly, husband's education, or parental education, employment status, occupation are important to consider. Besides, mother's age at first birth is important.

Response: 

We thank the reviewer for this comment. We have added paternal education to the model as a potential confounder. We found that it was significantly associated with delivery care but not with early neonatal mortality. Based on best practice, our preference is to include only the minimally sufficient adjustment set and therefore only include those variables that you hypothesize are associated with both your exposure and your outcome and not on the causal pathway (Reference: Kenneth J. Rothman, Sander Greenland, Timothy L. Lash. Modern Epidemiology, 3rd edition). Given that paternal education was not a strong confounder of the relationship between delivery care and early neonatal mortality, and that the mechanisms through which employment status and occupation might serve as a confounder are similar to those for paternal education, we decided not to add these additional paternal characteristics to the model. The model includes mother’s age at birth and birth order. Mother’s age at first birth will be correlated with these variables, especially as 40% of the sample are first births for which mother’s age at the time of the index birth will be the same as mother’s age at first birth so we decided not to add mother’s age at first birth to the model.

2.Why 2018 BDHS data has not been used for this study instead of BDHS 2014 data?

Response: BDHS 2018 data was published on December 14, 2020, after submission of this manuscript to PLOS One on September 28, 2020. We think that this analysis of the BDHS 2014 data provides a valuable contribution to the literature on delivery and newborn care and ENNM in Bangladesh in the MDG period and provides a foundation to build on with further analysis of the BDHS 2017-18 data to explore whether and how these relationships have changed as facility delivery has increased in Bangladesh. Therefore, we will explore replicating the analysis with the BDHS 2017-18 data in a separate paper.

3. How about the death rate among the missing cases? Total 309 cases have been omitted. Just in case if considerable amount of death rate is found among 309 cases, then suitable imputation method could be used to impute the missing observations.

Response: The 309 cases were in fact dropped because they were not the most recent live birth and therefore no data were collected in the DHS on ANC, essential newborn care and PNC. We have clarified that the sample includes only the most recent live births in the revised text on p7, lines 1-5. 

4. Have the twin babies considered here. If so then, whether it could be an important factor in the model.

Response: Thank you for highlighting this important point. The sample includes the youngest in a multiple birth when that is the most recent live birth. We have clarified this in the text on p. 7 lines 1-4 and have added an indicator variable for twin birth in the model. 

5. In Statistical Analysis section, Variance inflation factors (VIF) were assessed to examine collinearity between variables before entering them into the multivariable models? What cut off point was used here?

Response: We have used cut off point 5 for VIF to assess multicollinearity. In this study, Our calculated VIF was 1.77. This information now appears on p9, lines 12. 

6. The confounding could be existed here. Particularly, ANC visits and ENNM might be under important consideration in making such unobserved confounding. What is the response of authors in this regards?

Response: We agree that ANC care is a potential confounder of the relationship between delivery care and early neonatal mortality, as shown in the DAG that we have added as an annex in the revised paper to show more clearly which confounders we have included in the model. We have included two indicators of ANC care: any ANC with a medically trained provided and number of components of ANC received, as indicators of the quantity and quality of ANC. Number of ANC visits is another dimension of ANC, but it will be related to the two indicators of ANC that we have included and we chose the number of components of ANC as a more comprehensive indicator of the quality of ANC care. 

7. Is simple map as displayed in Figure 1 important? Rather authors could have mapping for Bangladesh by disagegating to division levels. Authors could use many software including ``mapReasy" R package (Islam et al., 2017).

Response: We have decided to drop Figure 1 in this paper.________________________________________

Additional Editor Comments:

1. In the abstract (and maybe elsewhere) it is not explicit who are “their counterparts”.

Response: We thank the reviewer for this comment. We have explained ‘their counterparts’ in the abstract (p1, line 17-18 – “compared to their counterparts who received 0-2 components of ENC”). Also, we specified the counterparts as those ‘who had not received any PNC’ on p1, lines 15-18.

2. Use multivariable instead of multivariate. To distinguish between these two terms, please refer this this https://academic.oup.com/ntr/advance-article/doi/10.1093/ntr/ntaa055/5812038

Response: We have corrected the term to multivariable analysis throughout. 

3. In descriptive results in the tables, clarify whether the frequencies and percentages are weighted or unweighted.

Response: We have added a sentence “All the tables were produced using survey-weighted percentages and counts” in the Statistical analysis section (p9, lines 18-19), 

4. It is unclear whether all the survey design features, including weighting and stratification, were taken into account.

Response: We have added a sentence “appropriate sampling weights for BDHS 2014 were applied using Stata’s survey estimation procedures (“svy” command) to get nationally representative estimates of the population of Bangladesh after adjusting for sample strata and clusters” in the Statistical analysis section (p9, lines 14-19).

5. P values of 0.000 should be written as <0.001

Response: We have made this correction in Tables 3a, 3b and 4. 

In Table 3, please present the results of the pre-specified exposure variables first.

Response: We have divided Table 3 into two tables in the revised version to respond to comments of reviewer 2. The primary exposure variable of interest in delivery in a health facility, which forms the columns in Tables 3a and 3b, consistent with the DAG that is added as an annex in the revision. The secondary exposure variables of interested are included in Table 3b. We have presented these primary and secondary exposure variables first in Table 4.

6. In presenting (and discussing) the results, please focus on the exposure variables of interest. This is because the multivariable model includes some variables that are on the causal pathway. For example, size of child at birth, a proxy for birthweight, is most likely a mediator between maternal socio-demographic characteristics and ENNM. Thus, the ORs for the socio-demographic characteristics may be biased.

Response: We would like to thank the review for this comment. We have rewritten the results section to focus on the exposure variables – see response to reviewer 2 comments 12-14 above. 

Journal Requirements:

Response: We have ensured that the manuscript meets PLOS ONE’s style requirements. 

2. In statistical methods, please clarify whether you corrected for multiple comparisons.

Response: We have not corrected for multiple comparisons in the analysis. In the revised paper we have focused on the significance of the primary and secondary exposure variables of interest so the number of hypotheses we focus on is relatively small and few variables are significant in our final models even at standard significance levels. We have added a sentence to clarify this in the Methods section on p.9 lines 17-18 as “Standard significance levels are reported (i.e., no correction for multiple comparisons).”

3. In your statistical analyses, please state whether you accounted for clustering by region/ state. For example, did you consider using multilevel models? If not, please provide your rationale for not doing so.

Response: All analysis was performed using survey weights and accounted for clustering. We have mentioned the appropriate sampling weights were applied using Stata’s survey estimation procedure (svy) to achieve nationally representative estimates of the population after adjusting for sample strata and clusters. [mentioned in p9, lines 14-19]. 

Response: We have included the ethics statement in the method section of the manuscript. 

5. We note that Figure 1 in your submission contain map images which may be copyrighted. All PLOS content is published under the Creative Commons Attribution License (CC BY 4.0), which means that the manuscript, images, and Supporting Information files will be freely available online, and any third party is permitted to access, download, copy, distribute, and use these materials in any way, even commercially, with proper attribution. For these reasons, we cannot publish previously copyrighted maps or satellite images created using proprietary data, such as Google software (Google Maps, Street View, and Earth). For more information, see our copyright guidelines: http://journals.plos.org/plosone/s/licenses-and-copyright.

5.1. You may seek permission from the original copyright holder of Figure 1 to publish the content specifically under the CC BY 4.0 license. 

5.2. If you are unable to obtain permission from the original copyright holder to publish these figures under the CC BY 4.0 license or if the copyright holder’s requirements are incompatible with the CC BY 4.0 license, please either i) remove the figure or ii) supply a replacement figure that complies with the CC BY 4.0 license. Please check copyright information on all replacement figures and update the figure caption with source information. If applicable, please specify in the figure caption text when a figure is similar but not identical to the original image and is therefore for illustrative purposes only.

Response: We have decided to drop Figure 1.

---

## [Decision Letter · Decision Letter 1]

13 Jul 2021

PONE-D-20-30510R1

The role of institutional delivery and newborn care in preventing early-neonatal mortality in Bangladesh

PLOS ONE

Dear Dr. Ijdi,

Thank you for submitting your manuscript to PLOS ONE. After careful consideration, we feel that it has merit but does not fully meet PLOS ONE’s publication criteria as it currently stands. Therefore, we invite you to submit a revised version of the manuscript that addresses the points raised during the review process.

Reviewers have raised valid concerns regarding data analysis and the dataset analyzed. Please consider analyzing the latest DHS data to provide the most updated information.

We look forward to receiving your revised manuscript.

Kind regards,

Calistus Wilunda, DrPH

Academic Editor

PLOS ONE

Reviewers' comments:

Reviewer's Responses to Questions

**Comments to the Author**

1. If the authors have adequately addressed your comments raised in a previous round of review and you feel that this manuscript is now acceptable for publication, you may indicate that here to bypass the “Comments to the Author” section, enter your conflict of interest statement in the “Confidential to Editor” section, and submit your "Accept" recommendation.

Reviewer #1: All comments have been addressed

Reviewer #2: (No Response)

Reviewer #3: (No Response)

2. Is the manuscript technically sound, and do the data support the conclusions?

Reviewer #1: Yes

Reviewer #2: No

Reviewer #3: Yes

3. Has the statistical analysis been performed appropriately and rigorously? 

Reviewer #1: Yes

Reviewer #2: No

Reviewer #3: Yes

4. Have the authors made all data underlying the findings in their manuscript fully available?

Reviewer #1: Yes

Reviewer #2: Yes

Reviewer #3: Yes

5. Is the manuscript presented in an intelligible fashion and written in standard English?

Reviewer #1: Yes

Reviewer #2: Yes

Reviewer #3: No

6. Review Comments to the Author

Reviewer #1: You may also include additional comments for the author, including concerns about dual publication, research ethics, or publication ethics.

No COmment

Reviewer #2: I'm repeating my comments that I submitted as an attachment:

The authors have made some significant changes. However, there are still many major issues that need to be addressed:

1. In response to aggregating 2/3 cycles of data to improve statistical power, the authors mentioned that the proportion of women delivering in health facilities has been increasing over time, affecting the relationship between delivery care and early neonatal mortality. If this is true, why would we believe findings from BDHS 2014 data would reflect the current scenario of the association? The BDHS 2017-18 dataset is already publicly available, and it should only take a couple of days to access the dataset. Then why would be we believe BDHS 2014 dataset is more likely to reflect the current scenario than BDHS 2017-18?

In Table 4 (model 3), the authors adjusted the model for many covariates. However, the number of events is only 69 in the study (Table 1). For example, there are only 69/29 ≈ 2 events per effective degrees of freedom. Therefore, the estimates from the current models could be unstable and invalid as per the events per variable (EPV) rule (https://doi.org/10.1016/0895-4356(95)00510-2; https://link.springer.com/article/10.1007/s10985-013-9290-4#ref-CR15;
https://doi.org/10.1093/aje/kwk052).

In the previous version, it was recommended to use an alternative method (e.g., propensity score full-matching, propensity score weighting) as a sensitivity analysis for the main findings. Since there are very few events and only 2 events per effective degrees of freedom (i.e., 2 EPV), why should we rely only on the logistic regression results? The odds ratio from the logistic regression also often suffers from many major problems (e.g., non-collapsible bias; https://doi.org/10.1177%2F0962280213505804). The authors should provide a strong rationale why they are insisted (i) using the BDHS 2014 dataset but not the BDHS 2017-18 dataset, (ii) relying only on results from a single model instead of sensitivity analysis of the main findings using an alternative model, and (iii) why EPV is not a major concern, or if EPV is a concern why not they consider multiple cycles to improve statistical power and stability of the model.

2. According to Annex 1, PNC and ENC are mediators in the relationship between place of delivery (exposure) and early neonatal mortality (outcome). Adjusting for a mediator leads to decompose the total effect into direct and indirect effects (https://doi.org/10.1093/ije/dyt127). Therefore, the authors should not adjust for PNC and ENC in the same model when exploring the total effect of delivery place on early neonatal mortality. To explore the effect of PNC and ENC on early neonatal mortality, they should run a separate model.

There are some minor issues as well:

1. The title needs to be changed. Considering the study design of BDHS, one can only explore the association of institutional delivery and newborn care with early-neonatal mortality. We need longitudinal studies to conclude whether institutional delivery and newborn care actually prevent early neonatal deaths. One example revised title is “Exploring the association of the place of delivery and newborn care with early-neonatal mortality in Bangladesh”. Otherwise, please justify why the term “preventing early-neonatal mortality” is appropriate in the title.

2. The study objective in the Introduction is written as “This study aims to explore the association of the place of delivery and newborn care with ENNM in Bangladesh.” However, the aim is different in the Abstract. Please restate and clearly specify the study objective in the Abstract.

3. Pleased drop the sentence from the abstract “Singleton births had 95% lower odds of dying in the first seven days of life compared to twin birth (aOR: 0.05; 95% CI: 0.01-0.16) in the same period.” This result is not aligned with the study’s objective. Instead, it is the Table 2 fallacy.

4. The conclusion in the Abstract is wrong. Since delivery at the health facility is not associated with early neonatal deaths, how the study findings highlight the importance of newborn and postnatal care in preventing early neonatal deaths? Please clarify.

5. The study’s strength “The sample size is also relatively large …” is not true. Considering the extremely rare event, the study’s sample size ~4000 is tiny (http://www.vanbelle.org/chapters/webchapter2.pdf;
https://doi.org/10.1016/0047-2352(86)90111-X). The authors should restate the argument or provide the rationale for using this as a strength.

Reviewer #3: I have gone through the revised manuscript and found insufficient responses to some vital points. Hence, still following questions must be responded properly to improve this manuscript.

1. Authors need to give more strong reference against their claim ``Based on best practice, our preference is to include only the minimally sufficient adjustment set" as mentioned in the answer of first point. Many studies found direct associations of parental education, employment status, occupation with neo-natal mortality. Otherwise, they need to carry out the analysis with these variables in the model.

2. The data of BDHS2018 is available and all the variable codes are same as BDHS2014. Authors must need to redo all analyses with this data. Authors mentioned that they will do the same analysis and make another paper which is pointless as the dataset is already available.

3. Authors did not mention whether there was no any missingness in the covariates considered apart from answering missingness among the discarded 309 cases.

4. Figure 1 in previous manuscript was suggested to revise by using MapReasy which is important to strengthen this manuscript but authors omitted this. Authors need to generate and include such graph.

5. Authors still mentioned in abstract and statistical analysis section that multivaiate logistic..... It should be multivaiable logistic....

6. Some references are missing pages, volume, series etc. E.g., references 18, 36, 41, 43.

7. Study strengths were unnecessarily been mentioned in the last section. Only limitations are OK.

8. Page 6, l12: DAG has been created and a minimally sufficient adjustment is set according to ref 27. There are many published well known studies but authors cited ref 27 which is not a published paper and not worthy. Here how had DAG been created?

9. Data source is insufficiently discussed.

7. PLOS authors have the option to publish the peer review history of their article (what does this mean?). If published, this will include your full peer review and any attached files.

Reviewer #1: No

Reviewer #2: No

Reviewer #3: No

---

## [Author Response · Author response to Decision Letter 1]

25 Aug 2021

Reviewer’s #2

I am repeating my comments that I submitted as an attachment:

The authors have made some significant changes. However, there are still many major issues that need to be addressed: 

1. In response to aggregating 2/3 cycles of data to improve statistical power, the authors mentioned that the proportion of women delivering in health facilities has been increasing over time, affecting the relationship between delivery care and early neonatal mortality. If this is true, why would we believe findings from BDHS 2014 data would reflect the current scenario of the association? The BDHS 2017-18 dataset is already publicly available, and it should only take a couple of days to access the dataset. Then why would be we believe BDHS 2014 dataset is more likely to reflect the current scenario than BDHS 2017-18? 

In Table 4 (model 3), the authors adjusted the model for many covariates. However, the number of events is only 69 in the study (Table 1). For example, there are only 69/29 ≈ 2 events per effective degrees of freedom. Therefore, the estimates from the current models could be unstable and invalid as per the events per variable (EPV) rule (https://doi.org/10.1016/0895-4356(95)00510-2; https://link.springer.com/article/10.1007/s10985-013-9290-4#ref-CR15;
https://doi.org/10.1093/aje/kwk052). 

In the previous version, it was recommended to use an alternative method (e.g., propensity score full-matching, propensity score weighting) as a sensitivity analysis for the main findings. Since there are very few events and only 2 events per effective degrees of freedom (i.e., 2 EPV), why should we rely only on the logistic regression results? The odds ratio from the logistic regression also often suffers from many major problems (e.g., non-collapsible bias; https://doi.org/10.1177%2F0962280213505804). The authors should provide a strong rationale why they are insisted (i) using the BDHS 2014 dataset but not the BDHS 2017-18 dataset, (ii) relying only on results from a single model instead of sensitivity analysis of the main findings using an alternative model, and (iii) why EPV is not a major concern, or if EPV is a concern why not they consider multiple cycles to improve statistical power and stability of the model. 

Answer: Our justification: We submitted our manuscript on 29 September 2020, roughly two and half months before the public release of the latest BDHS (i.e., 2017-18 round) data. We received our first round of review on 14 January 2021, which recommended redoing the analysis using the latest BDHS round. We respectfully disagreed with that comment, mentioning that re-analyzing with the BDHS 2017-18 data fell outside the scope of the manuscript. Given the timing of our initial submission and the amount of effort that has been put to prepare the manuscript, we would like to re-iterate our position regarding this manuscript. We do not believe that the data from 2014 have become outdated, as a Google Scholar search indicates that nearly 20 articles were published in the last one year using these data, including one published in PLOS One in July 2021 (doi: 10.1371.journal.pone.0254777). 

We thank the PLOS ONE for accepting our justification to use the 2014 BDHS data for this manuscript. 

2. According to Annex 1, PNC and ENC are mediators in the relationship between place of delivery (exposure) and early neonatal mortality (outcome). Adjusting for a mediator leads to decompose the total effect into direct and indirect effects (https://doi.org/10.1093/ije/dyt127). Therefore, the authors should not adjust for PNC and ENC in the same model when exploring the total effect of delivery place on early neonatal mortality. To explore the effect of PNC and ENC on early neonatal mortality, they should run a separate model. 

Answer: We would like to thank the reviewers for this comment. We have modified the DAG accordingly; please see the updated DAG in Supplementary Information “S1 Figure”.

There are some minor issues as well: 

1. The title needs to be changed. Considering the study design of BDHS, one can only explore the association of institutional delivery and newborn care with early-neonatal mortality. We need longitudinal studies to conclude whether institutional delivery and newborn care actually prevent early neonatal deaths. One example revised title is “Exploring the association of the place of delivery and newborn care with early-neonatal mortality in Bangladesh”. Otherwise, please justify why the term “preventing early-neonatal mortality” is appropriate in the title. 

Answer: We would like to thank the reviewer for this comment. We have changed the title of the paper as “Exploring association between place of delivery and newborn care with early-neonatal mortality in Bangladesh”.

2. The study objective in the Introduction is written as “This study aims to explore the association of the place of delivery and newborn care with ENNM in Bangladesh.” However, the aim is different in the Abstract. Please restate and clearly specify the study objective in the Abstract. 

Answer: We thank the reviewer for this comment. The abstract has been updated accordingly, as “The purpose of this study is to explore the association between place of delivery and newborn care with early neonatal mortality (ENNM)”, in page 1, lines 4-6.

3. Pleased drop the sentence from the abstract “Singleton births had 95% lower odds of dying in the first seven days of life compared to twin birth (aOR: 0.05; 95% CI: 0.01-0.16) in the same period.” This result is not aligned with the study’s objective. Instead, it is the Table 2 fallacy. 

Answer: We would like to thank the reviewer for this comment. We have dropped the sentence “Singleton births ….. in the same period” from the abstract.

4. The conclusion in the Abstract is wrong. Since delivery at the health facility is not associated with early neonatal deaths, how the study findings highlight the importance of newborn and postnatal care in preventing early neonatal deaths? Please clarify. 

Answer: We have modified the conclusion accordingly, in page 2, lines 2-6, as “Further, findings suggest that increasing the proportion of women who give birth in a healthcare facility is not sufficient to reduce ENNM by itself; to realize the theoretical potential of facility delivery to avert neonatal deaths, we must also ensure quality of care during delivery, guarantee all components of ENC, and provide high-quality early PNC.”

5. The study’s strength “The sample size is also relatively large …” is not true. Considering the extremely rare event, the study’s sample size ~4000 is tiny (http://www.vanbelle.org/chapters/webchapter2.pdf;
https://doi.org/10.1016/0047-2352(86)90111-X). The authors should restate the argument or provide the rationale for using this as a strength. 

Answer: We would like to thank the reviewer for this comment. We have restated the argument as a limitation at page 22, lines 7-9. 

Reviewer # 3

I have gone through the revised manuscript and found insufficient responses to some vital points. Hence, still following questions must be responded properly to improve this manuscript.

1. Authors need to give more strong reference against their claim “Based on best practice, our reference is to include only the minimally sufficient adjustment set" as mentioned in the answer of first point. Many studies found direct associations of parental education, employment status, occupation with neo-natal mortality.

Otherwise, they need to carry out the analysis with these variables in the model.

Answer: As we have mentioned, in our first revision we have already included paternal education. As a reminder, we define a variable as a confounder if it is associated with both the exposure and the outcome and not on the causal pathway, as is best practice in the field of epidemiology. Because we do not believe that paternal employment status and occupation are associated with place of delivery, ENC, or PNC, we cannot include these variables in our regression model. 

2. The data of BDHS2018 is available and all the variable codes are same as BDHS2014. Authors must need to redo all analyses with this data. Authors mentioned that they will do the same analysis and make another paper which is pointless as the dataset is already available.

Answer: According to the editor’s decision we have decided to keep the analysis of the paper using BDHS 2014 data. 

3. Authors did not mention whether there was no any missingness in the covariates considered apart from answering missingness among the discarded 309 cases.

Answer: In the first round of review, regarding missing values we have already mentioned that “The 309 cases were in fact dropped because they were not the most recent live birth, thus ANC, PNC information were not available.”

4. Figure 1 in previous manuscript was suggested to revise by using MapReasy which is important to strengthen this manuscript but authors omitted this. Authors need to generate and include such graph.

Answer: We have dropped the map (Figure 1) according to editor’s comment in the first round of review.

5. Authors still mentioned in abstract and statisticcal analysis section that multivariate logistic..... It should be multivaiable logistic....

Answer: We have carefully checked the manuscript and we do not find any remaining use of the term ‘multivariate logistic.’ 

6. Some references are missing pages, volume, series etc. E.g., references 18, 36, 41, 43.

Answer: We would like to thank the reviewer for the very careful review. We have checked and confirming that references 18, 36, 41 are properly cited. We have added the volume number and electronic identifier (uses by PLOS ONE instead of page numbers) for reference # 43. 

7. Study strengths were unnecessarily been mentioned in the last section. Only limitations are OK.

Answer: We would like to thank the reviewer for this comment. We have decided to keep both strengths and limitations in this manuscript.

8. Page 6, l12: DAG has been created and a minimally sufficient adjustment is set according to ref 27. There are many published well known studies but authors cited ref 27 which is not a published paper and not worthy. Here how had DAG been created?

Answer: We have referred a textbook of epidemiology, which we found appropriate here. Respectfully, Drs. Kenneth Rothman, Sander Greenland, and Tim Lash and the content of their book, Modern Epidemiology, are more than ‘worthy’ to be cited to justify use of a DAG to select a minimally sufficient adjustment set. 

9. Data source is insufficiently discussed.

Answer: We have specified the data source in page 6, lines 16-18, as “The study uses data from the 2014 Bangladesh Demographic and Health Survey (BDHS)- a nationally representative cross-sectional survey using a two-stage stratified random sampling of households, which covered all districts and administrative divisions of Bangladesh.”

Editor’s comment:

Reviewers have raised valid concerns regarding data analysis and the dataset analyzed. Please consider analyzing the latest DHS data to provide the most updated information. 

Answer: According to the editions mail (dated August 19, 2021) mentioned “the use of the 2014 dataset is justified given that the manuscript was submitted before the public release of the latest BDHS data.” Thus, we are keeping the analysis in this paper using the BDHS 2014 data.

---

## [Decision Letter · Decision Letter 2]

24 Dec 2021

Exploring association between place of delivery and newborn care with early-neonatal mortality in Bangladesh

PONE-D-20-30510R2

Dear Dr. Ijdi,

We’re pleased to inform you that your manuscript has been judged scientifically suitable for publication and will be formally accepted for publication once it meets all outstanding technical requirements.

Kind regards,

Calistus Wilunda, DrPH

Academic Editor

PLOS ONE

Additional Editor Comments (optional):

Reviewers' comments:

Reviewer's Responses to Questions

**Comments to the Author**

1. If the authors have adequately addressed your comments raised in a previous round of review and you feel that this manuscript is now acceptable for publication, you may indicate that here to bypass the “Comments to the Author” section, enter your conflict of interest statement in the “Confidential to Editor” section, and submit your "Accept" recommendation.

Reviewer #1: All comments have been addressed

Reviewer #2: All comments have been addressed

2. Is the manuscript technically sound, and do the data support the conclusions?

Reviewer #1: Yes

Reviewer #2: Yes

3. Has the statistical analysis been performed appropriately and rigorously? 

Reviewer #1: Yes

Reviewer #2: Yes

4. Have the authors made all data underlying the findings in their manuscript fully available?

Reviewer #1: Yes

Reviewer #2: Yes

5. Is the manuscript presented in an intelligible fashion and written in standard English?

Reviewer #1: Yes

Reviewer #2: Yes

6. Review Comments to the Author

Reviewer #1: revision: Similar research has been published: "Effect of antenatal care and social wellbeing on early neonatal mortality in

Bangladesh" using MICS data. this findings may be compare in the discussion section

Reviewer #2: The authors addressed most of the comments I had. I had no more comments except for the sensitivity analysis for the main findings. As explained in the previous version, the statistical power and stability of the model can be problematic. The reason is obvious since there are only 2 EPV. Otherwise, the paper is in good shape and can be accepted for publication. Congratulations to the authors!

7. PLOS authors have the option to publish the peer review history of their article (what does this mean?). If published, this will include your full peer review and any attached files.

Reviewer #1: No

Reviewer #2: No

---

## [Editor Report · Acceptance letter]

13 Jan 2022

PONE-D-20-30510R2 

Exploring association between place of delivery and newborn care with early-neonatal mortality in Bangladesh 

Dear Dr. Ijdi:

I'm pleased to inform you that your manuscript has been deemed suitable for publication in PLOS ONE. Congratulations! Your manuscript is now with our production department. 

Kind regards, 

on behalf of

Dr. Calistus Wilunda 

Academic Editor

PLOS ONE